

**Aerosol-meteorology feedback diminishes the trans-boundary**
**transport of black carbon into the Tibetan Plateau**
Yuling Hu[a], Shichang Kang[b, c, *], Haipeng Yu[a], Junhua Yang[b], Mukesh Rai[b], Xiufeng
Yin[b], Xintong Chen[b], Pengfei Chen[b]
[a]Key Laboratory of Land Surface Process and Climate Change in Cold and Arid
Regions, Northwest Institute of Eco-Environment and Resources, Chinese Academy of
Sciences, Lanzhou 730000, China
[b]State Key Laboratory of Cryospheric Science, Northwest Institute of Eco-Environment
and Resources, Chinese Academy of Sciences, Lanzhou 730000, China
[c]University of Chinese Academy of Sciences, Beijing 100049, China
Correspondence to: Shichang Kang (shichang.kang@lzb.ac.cn)
**Abstract**
Black carbon (BC) exerts potential effect on climate, especially in the Tibetan Plateau
(TP), where the cryosphere and environment are very sensitive to climate change. The
TP saw the record-breaking aerosol pollution event during the period from April 20 to
May 10, 2016. This paper investigated the meteorological causes, trans-boundary
transport flux of BC, and aerosol-meteorology feedback as well as its effect on trans-
boundary transport flux of BC during this severe aerosol pollution event via using
observational and reanalysis dataset and simulation from the coupled meteorology and
aerosol/chemistry model (WRF-Chem). By analyzing the weather maps derived from
reanalysis dataset, it is found that the plateau vortex and southerly winds were key



factors that contributed to the severe aerosol pollution event. Subsequently, with the
good performance of WRF-Chem model on the spatiotemporal characteristics of
meteorological conditions and aerosols, the trans-boundary transport flux of BC during
the pollution event was investigated. The results show that the vertically integrated
cross-Himalayan transport flux of BC decreases from west to east, with the largest
transport flux of 20.8 mg m$^{-2}$ s$^{-1}$ occurring at the deepest mountain valley in
southwestern TP. Results from simulations with and without aerosol-meteorology
feedback show that aerosols induce significant changes in meteorological conditions in
the southern TP and Indo-Gangetic Plain (IGP), with the atmospheric stratification
being more stable and the planetary boundary layer height decreasing in both regions,
and 10-m wind speed increasing in the southern TP but decreasing in the IGP. Changes
in meteorological conditions in turn lead to a decrease of surface BC concentration with
value up to 0.16 μg/m$^3$ (50%) in the southern TP and an increase of surface BC
concentration with value up to 2.2 μg/m$^3$ (75%) in the IGP. By excavating the impact
of aerosol-meteorology feedback on the trans-boundary transport flux of BC, it has been
acquired that the aerosol-meteorology feedback decreases the integrated transport flux
of BC from central and western Himalayas towards the TP. This study not only provides
crucial policy implications for mitigating glacier melt caused by aerosols over the TP,
but also is of great significance to the ecological environment protection for the TP.
**Keywords**: Tibetan Plateau; Black carbon; Transport flux; Aerosol-meteorology
feedback; WRF-Chem



## 1 Introduction

Known as "the Third Pole", the Tibetan Plateau (TP) plays a significant role in driving the climate change in the Northern Hemisphere and even the globe through thermal and dynamical forcing (Lau et al., 2006;Wu et al., 2007). What's more, with the concentrated glacier and snow cover outside of the polar regions, the TP supplies a substantial portion of the water demand for almost 2 billion people (Yao et al., 2022). However, in recent years, numerous studies have reported that the TP experienced significant and rapid climate warming during the last few decades (Kang et al., 2010;You et al., 2016;You et al., 2021). As a result of this intensive warming, the glaciers over the TP have undergone unprecedented widespread losses and accelerated retreats (Kang et al., 2010;Yao et al., 2007). Besides high levels of greenhouse gases (Duan et al., 2006), other factors like atmospheric heating and snow albedo reduction due to absorbing aerosols also contribute a large portion to this climate warming and glacier retreat (Xu et al., 2009;Zhang et al., 2021;Kang et al., 2019b). However, with an average elevation exceeding 4 km, the TP is relatively undisturbed by human activities and is one of the most pristine regions on the earth. Hence, aerosols over the TP are mainly sourced from its surrounding regions (Kang et al., 2019b). Particularly, with nearly half of the world's population and heavy industry, South and East Asia adjacent to the TP are the world's hotspots for aerosol pollution (Lelieveld et al., 2016). Driven by atmospheric circulation, aerosols over South and East Asia can be transported to the TP, and then exert striking effect on hydrological cycle and climate (Wu et al., 2008;Ramanathan et al., 2005;Liu et al., 2014). Previous studies indicated



that aerosols over the TP are primarily transported via the typical long-distance trans-
boundary transport event (Kang et al., 2019a). It is therefore paramount to excavate the
meteorological causes of the severe aerosol pollution event as well as the trans-
boundary transport flux of aerosols during the severe aerosol pollution event.

Black carbon (BC) exerts substantial impact on climate through several

mechanisms, including heating the atmosphere by absorbing shortwave and longwave
radiation, darkening the surface of snow and ice and accelerating the melt of cryosphere,
and modifying the optical and microphysical properties of clouds (Kang et al.,
2019b;Ramanathan and Carmichael, 2008;Flanner et al., 2007;Skiles et al., 2018).
Estimation from the literature shows that BC is the second most important type of
human forcing after carbon dioxide, with a globally climate forcing of 1.2 W m$^{-2}$
(Ramanathan and Carmichael, 2008;Chung et al., 2005). Moreover, the radiative
forcing of BC in snow and ice is approximately twice as high as that of carbon dioxide
and other types of anthropogenic forcing (Flanner et al., 2007;Qian et al., 2011;Hansen
and Nazarenko, 2004). Particularly, as a sensitive area to global climate change, the TP
has seen an increase in BC content in recent years (Xu et al., 2009). It is evident that
BC plays a substantial role in the climate and environmental change over the TP.
However, previous studies primarily focused on the origin of BC and its climatic effect
over the TP on annual and seasonal timescales (Yang et al., 2018;Hu et al., 2022;Rai et
al., 2022). With respect to the trans-boundary transport flux of BC towards the TP
during the severe aerosol pollution event on the synoptic scale, there is still a blank,
which should be studied urgently.



88  In addition, severe aerosol pollution events are usually accompanied by complex

89 aerosol-meteorology feedback. Moreover, numerous studies have revealed that the

90 aerosol-meteorology feedback has substantial effect on surface aerosol concentration

91 (Wu et al., 2019;Zhang et al., 2018;Zhao et al., 2017;Hong et al., 2020;Chen et al.,

92 2019a;Gao et al., 2015a). For instance, some studies analyzed the aerosol-meteorology

93 feedback and its effect on surface $PM_{2.5}$ concentration during heavy aerosol pollution

94 events in winter in northern China, and found that positive aerosol-meteorology

95 feedback can increase the surface $PM_{2.5}$ concentration (Li et al., 2020;Qiu et al.,

96 2017;Wu et al., 2019). Nonetheless, some other studies suggested that the aerosol-

97 meteorology feedback can reduce the surface $PM_{2.5}$ concentration in Beijing (Gao et al.,

98 2015a;Zheng et al., 2015). Based on the in-situ observational data, Zhong et al., (2018)

99 analyzed the aerosol-meteorology feedback during several air pollution events in

100 Beijing and indicated that 70% of the increase in surface $PM_{2.5}$ concentration in the

101 cumulative outbreak stage of haze could be attributed to the aerosol-meteorology

102 feedback. The above mentioned studies are mainly focused on the economically

103 developed central and eastern China; however, the TP, which has a very high altitude

104 and complex topography along with tough environment and scarce in-situ observational

105 data, systematic and comprehensive studies on aerosol-meteorology feedback and its

106 effect on surface BC concentration are still lacking, which need urgent investigation.

107 Furthermore, what effect the aerosol-meteorology feedback having on the trans-

108 boundary transport flux of BC also remains unclear, which is also worthy in-depth study.

109 Therefore, in this study, we attempt to investigate the meteorological causes, the trans-



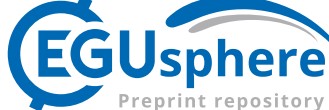

boundary transport flux of BC, and the aerosol-meteorology feedback as well as its
effect on the trans-boundary transport flux of BC during one severe aerosol pollution
event by using observational and reanalysis dataset, and numerical simulation with the
advanced regional climate-chemistry model, the Weather Research and Forecasting
with Chemistry (WRF-Chem). This study not only provides crucial policy implications
for mitigating glacier melt caused by aerosols over the TP, but also is of great
significance to the ecological environment protection for the TP. This paper is
organized as follows. Section 2 introduces data used in this study, the definition of
aerosol pollution event, WRF-Chem model, and experimental details. Section 3
investigates the meteorological causes, the trans-boundary transport flux of BC, and the
aerosol-meteorology feedback as well as its effect on transport flux of BC during the
severe aerosol pollution event. Section 4 presents the main conclusions.

## 2. Data, Definition of aerosol pollution event, and WRF-Chem model and

*2.1 Data*

*2.1.1 ERA-Interim*

To explore the meteorological causes of the severe aerosol pollution event, the
geopotential height, air temperature (T), and wind fields at 500 hPa with a horizontal
resolution of 0.05°×0.05° during the period from April 20 to May 10, 2016 are from the
European Center for Medium-Range Weather Forecasts interim reanalysis (ERA-
Interim). To evaluate the model performance on meteorology, 2-m air temperature (T2),



2-m dew point temperature, 10-m wind speed (U10), and wind fields at 500 hPa with a
horizontal resolution of 0.05°×0.05° are also obtained from ERA-Interim. It should be
noted that 2-m relative humidity (RH2) used to validate model performance is
calculated by 2-m dew point temperature and T2.
*2.1.2 AERONET*
The identification of aerosol pollution events on the TP is based on quality-assured
data from the Aerosol Robotic Network (AERONET), which was established by the
U.S. National Aeronautics and Space Administration (NASA) (Holben et al., 1998) and
is used to retrieve aerosol properties via sun photometers (Dubovik and King, 2000).
AERONET data, including instantaneous data and daily average by calculating the
diurnal average of the instantaneous values (Holben et al., 1998), are available at three
levels: level 1.0 (unscreened), level 1.5 (cloud-screened), and level 2.0 (cloud-screened
and quality assured data) (Smirnov et al., 2009). In this paper, the aerosol optical depth
(AOD) and fine-mode AOD at a standard wavelength of 500 nm used to define the
aerosol pollution event are based on the Spectral Deconvolution Algorithm (SDA)
version 3, level 2.0 (O'Neill et al., 2003;O'Neill et al., 2008). In addition, this kind of
AOD data was also used to verify the model performance on the temporal variation of
AOD at different sites over the study area.
*2.1.3 MODIS*
The Moderate Resolution Imaging Spectroradiometer (MODIS) instrument
aboard the Terra and Aqua satellites are designed with 36 spectral bands ranging from



0.4 to 15 μm and a high spatial resolution for retrieving reliable and extensive
information about solar radiation, atmosphere, ocean, cryosphere, and land. The
enhanced Deep Blue aerosol retrieval algorithm has substantially improved the
collection 6 aerosol product over the entire land region, especially in deserts and urban
regions (Hsu et al., 2013). Herein, to verify the model performance on the spatial
distribution of AOD, the AODs based on the Deep Blue algorithm at 550 nm with a
horizontal resolution of 1°×1° and a daily temporal resolution from MODIS/Aqua
Level-3 collection 6 products during the period from April 20 to May 10, 2016 are used.
It should be noted that MODIS onboard the Aqua satellite passes over the equator at
13:30 local time.
*2.1.4 MERRA-2*
The second Modern-Era Retrospective analysis for Research and Applications
(MERRA-2), which is introduced to replace the original MERRA reanalysis because of
the advances in the Goddard Earth Observing System Model, Version 5 (GEOS-5) data
assimilation system, is a NASA atmospheric reanalysis, beginning in 1980 (Gelaro et
al., 2017). MERRA-2 is the first long-term global reanalysis to assimilate space-based
observations of aerosols, including assimilation of AOD retrieved from the Advanced
Very High Resolution Radiometer instrument over the oceans (Heidinger et al., 2014),
the MODIS (Levy et al., 2010), non-bias-corrected AOD retrieved from the Multiangle
Imaging SpectroRadiometer (Kahn et al., 2005) over bright surfaces, and ground-based
AERONET observations (Holben et al., 1998). This dataset includes all the processes



of aerosol transport, deposition, microphysics, and radiative forcing and has
considerable skill in showing numerous observable aerosol properties (Gelaro et al.,
2017;Randles et al., 2017), including dust, sulfate, organic carbon, BC, and sea salt
aerosols (Chin et al., 2002;Colarco et al., 2010). As the first long-term aerosol
reanalysis dataset, MERRA-2 has been adequately evaluated in previous studies
(Buchard et al., 2017;Che et al., 2019;Sun et al., 2019). In this paper, the hourly surface
BC concentration ($kg/m^3$), which has a spatial resolution of longitude-by-latitude grid
of approximately 0.625°×0.5°, is used to validate the model performance on BC.
*2.2 Definition of aerosol pollution event*

The two main reference sites used in this study are Nam Co Monitoring and

Research Station for Multisphere Interactions (Nam Co), situated in inland TP
(30.77 °N, 90.96 °E, 4730 m a.s.l.), and Qomolangma Station for Atmospheric and
Environmental Observation and Research (QOMS, 28.36 °N, 86.95 °E, 4276 m a.s.l.),
located on the northern slope of Mt. Everest in the central Himalayas. The Nam Co and
QOMS stations joined the AERONET network in 2006 and 2009, respectively, and are
continuously functioning up to date. Both stations are background stations with less
human activities and can be regarded as a representative site for the inland of TP and
the southern TP, respectively (Pokharel et al., 2019). Figure S1 in the supporting
information (SI) shows daily mean of AOD at a standard wavelength of 500 nm lying
above the 95th percentile at Nam Co and QOMS stations since 2006 and 2009,
respectively. As can be seen, the observed most severe aerosol pollution event ever
recorded in the remote TP occurred during the period from April 27 to May 4, 2016,




persisting at least eight days simultaneously at Nam Co and QOMS sites. Figure 1
presents the temporal variation in daily mean AOD and fine-mode AOD at 500 nm for
both stations during the period from April 20 to May 10, 2016. Notably, the changes in
daily mean AOD and fine-mode AOD are synchronized and the value of daily mean
fine-mode AOD is very close to that of daily mean AOD, indicating that the fine
particulate matter dominated this severe aerosol pollution event. Meanwhile, it is also
acquired from Figure 1 that the most polluted period during this pollution event is from
April 27 to May 4. Specifically, at Nam Co station, the observed highest daily mean
AOD and fine-mode AOD with values of 0.65 and 0.64 appeared on April 29, 2016,
while at QOMS station, the measured highest daily mean AOD and fine-mode AOD
with values of 0.42 and 0.39 occurred on May 1, 2016. According to the previous study,
the baseline values of AOD at Nam Co and QOMS are 0.029 and 0.027, respectively
(Pokharel et al., 2019). Thus, the observed highest AOD at Nam Co and QOMS stations
during this severe aerosol pollution event is at least one order of magnitude than that of
baseline.

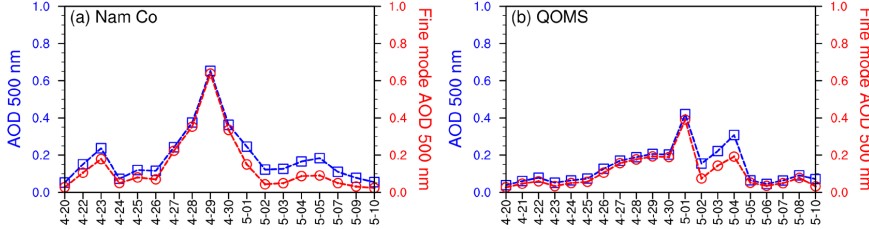


Figure 1 Time series of daily mean AOD (blue) and fine-mode AOD (red) at 500 nm at
Nam Co (a) and QOMS (b) stations during the period from April 20 to May 10, 2016.





*2.3 WRF-Chem Model Configuration, Experimental Design and Emissions*
The WRF-Chem model is a fully coupled regional dynamical/chemical transport
model that considers gas-phase chemistry, photolysis, and aerosol mechanism (Grell et
al., 2005). The model can simulate the emission, transport, mixing, chemical reactions,
and deposition of trace gases and aerosol simultaneously with the meteorological fields.
It has been successfully applied in air pollution studies over the TP and adjacent regions
(Yang et al., 2018;Chen et al., 2018;Hu et al., 2022). The version used in this study is
based on v3.9.1. As shown in Figure 2, the simulation domain is centered at 31°N,
87.5 °E, covering the whole TP and its surroundings. The model simulations are
conducted at a 15-km horizontal resolution using a Lambert conformal mapping with
259 (west-east) × 179 (north-south) grid cells. There are 30 vertical sigma levels for all
grids, extending from the surface to 50 hPa. The key physical parameterization options
used in this study include the Noah land surface model (Ek et al., 2003;Chen et al.,
2010) and the Monin-Obukhov scheme for the surface layer physical processes
(Srivastava and Sharan, 2017), the double-moment Morrison microphysical
parameterization (Morrison et al., 2009) with the Grell-Freitas (GF) cumulus scheme
(Grell and Freitas, 2014), the Mellor-Yamada-Janjic (MYJ) planetary boundary layer
scheme with local vertical mixing (Janjić, 1994), and the Rapid Radiative Transfer
Model for General circulation models (RRTMG) coupled with the aerosol radiative
effect for both longwave and shortwave radiation (Iacono et al., 2008). In respect to the
chemical parameterization options, the Carbon-Bond Mechanism version Z (CBMZ)
gas-phase chemistry mechanism (Zaveri and Peters, 1999) combined with the Model



for Simulating Aerosol Interactions and Chemistry (MOSAIC) aerosol module (Zaveri
et al., 2008) was chosen for aerosol simulation. The MOSAIC aerosol scheme uses an
approach of segmentation to represent aerosol size distribution with four or eight
discrete size bins (Fast et al., 2006). In this paper, the aerosol size is divided into four
bins. Aerosol species simulated by MOSAIC scheme include sulfate, methanesulfonate,
nitrate, chloride, carbonate, ammonium, sodium, calcium, BC, primary organic mass,
liquid water, and other inorganic mass.
The initial and boundary conditions for meteorological fields are obtained from
the National Centers for Environmental Prediction (NCEP) Final Analysis (FNL) data
with a 1° × 1° spatial resolution and a 6-h temporal resolution
(https://rda.ucar.edu/datasets/ds083.2/). Anthropogenic emissions, such as CO, VOCs,
NOx, NH$_3$, BC, OC, SO$_2$, PM$_{2.5}$, and PM$_{10}$, are taken from the Emission Database for
Global Atmospheric Research (EDGAR)-Hemispheric Transport Air Pollution version
2 (HTAPv2) emission inventory (https://edgar.jrc.ec.europa.eu/dataset_htap_v2) for the
year 2010. Detailed information on the HTAP inventory can be found in Janssens-
Maenhout et al. (2015). The biogenic emissions are based on the Model of Emissions
of Gases and Aerosols from Nature (MEGAN) (Guenther et al., 2006;Guenther et al.,
2012), and the biomass burning emissions were calculated with the high resolution fire
emissions based on the Fire INventory from NCAR (FINN) (Wiedinmyer et al., 2011) .
In addition, the mozbc utility and the Community Atmosphere Model with
Chemistry (CAM-chem) (Buchholz et al., 2019) dataset are used to create improved
chemical initial and boundary conditions. The simulation is conducted from April 10,





2016 to May 10, 2016, and the first ten days is used for model spin-up. The results from
April 20, 2016 to May 10, 2016 are used for the analysis.

A total of two experiments are designed in this study, one is a control experiment

(CONT) and the other is a sensitive experiment (SEN). In the CONT, the simulation is
conducted using WRF-Chem model with aerosol-meteorology feedback turned on. The
SEN is exactly the same as CONT except that the feedback between aerosol and
meteorology is turned off. The difference between CONT and SEN is considered as
effect induced by aerosol-meteorology feedback.

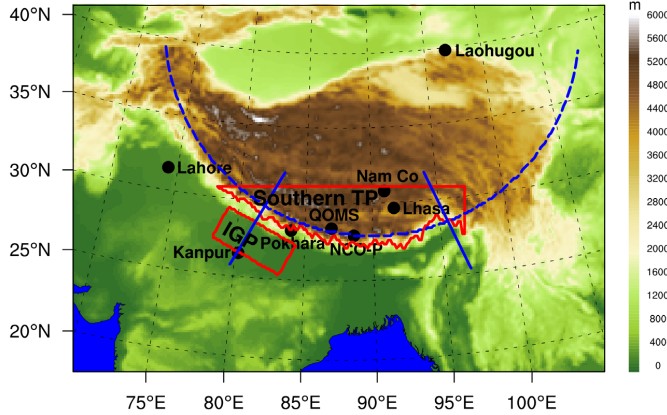


Figure 2 WRF-Chem model domain and terrain (shading; m). Black solid dots indicate
stations used to verify model performance on meteorological conditions and aerosols.
The solid red lines and its inner area denote the southern TP and Indo-Gangetic Plain.
The blue dashed line and two solid lines represent the cross sections for analysis in the
following.



## 3 Results and discussion

*3.1 Meteorological causes of the severe aerosol pollution event*

Excessive emissions and adverse meteorological conditions are the two most important factors influencing air quality (Zhang et al., 2019;Wang et al., 2019;Chen et al., 2019b;Liu et al., 2017). The TP, which has a small population density and a low degree of industrialization, is one of the most pristine regions on the earth. Moreover, as mentioned above, AOD at background stations of Nam Co and QOMS with less human activity is significantly higher than that of baseline from April 27 to May 4, 2016, with the highest value at least one order of magnitude than that of baseline. Therefore, it can be inferred that aerosols over the TP during this severe aerosol pollution event are mainly sourced from adjacent regions by long-range transport, which is consistent with the results reported in a previous study (Kang et al., 2019b). In fact, atmospheric circulation, as the main driving force of atmospheric aerosols, plays a substantial role in the long-range transport of aerosols. Therefore, with little change in emission source, analyzing the meteorological conditions during the severe pollution event is very crucial. Figure 3 shows weather maps at 500 hPa based on ERA-Interim reanalysis dataset. It is found that, during 08:00–14:00 Beijing Time (hereafter BJT) on April 27, straight westerly airflow prevailed at 500 hPa over the TP (Figure 3a), which transported aerosols from northwestern South Asia to the TP. Subsequently, wind field shear occurred over the plateau at 20:00 on April 27 (Figure 3b), and plateau vortex generated at 02:00 on April 28 (Figure 3c), which is conducive to the accumulation of aerosols in the inland of the plateau. From 08:00 on April 28 to 08:00 on April 29, the



plateau vortex stabilized over the TP, and the aerosol concentrations at Nam Co station
continued increasing (figure not shown). At 14:00 on April 29, the plateau vortex
disappeared (Figure 3d) and the aerosol concentrations peaked at Nam Co station.
Meanwhile, a high-pressure system was located to the west side of the plateau
accompanied by a trough in the foreside (Figure 3d). Thus, southwesterly airflow in
front of the trough transported aerosols from northern South Asia to the southern TP
(Figure 3e). As a result, the aerosol concentrations at QOMS station increased and
peaked on May 1. At 14:00 on May 1, the high-pressure system moved eastward from
the west side to the inland of the TP and northerly winds ahead this high-pressure
system prevailed over the TP (Figure 3f), which wafted aerosols away from the TP and
aerosol concentrations at QOMS station began to decrease.

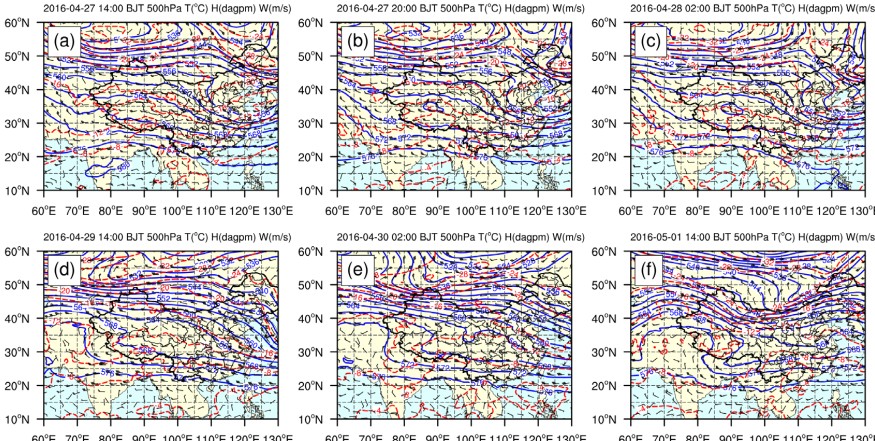


Figure 3 Weather maps at 500 hPa over the study area during the severe aerosol
pollution event based on ERA-Interim. The blue lines are isopleths of geopotential
height (unit: dagpm), the red lines are isotherms (unit: °C), and wind speed (unit: m/s)
and direction were denoted by wind barb.




*3.2 Evaluating model performance on meteorology and chemistry*

*3.2.1 Validation of model performance on meteorology*

Validating model performance on meteorology is critical for assuring accuracy in
simulating aerosol concentrations. This is because meteorological conditions are
closely associated with aerosol growth, transport, and deposition. Herein, to validate
the model performance on meteorological conditions, the temporal variation in the
simulated and reanalyzed T2, RH2, and U10 at Nam Co, QOMS, Kanpur, and Lahore
stations are shown in Figure S2 in the SI. The model reasonably represents the correct
temporal trend of T2 at four stations although underestimations are detected at Nam Co
and QOMS stations. The variation trend of RH2 from simulation is in high consistent
with that from reanalysis at Kanpur and Lahore stations. However, at Nam Co and
QOMS stations, a relatively larger discrepancy is observed between simulation and
reanalysis, which might be related to the high altitude and complex topography there.
For U10, the simulated trend on average coincides with the observed trend at Nam Co,
Kanpur, and Lahore stations. The corresponding statistics, including sample size (N),
observed mean, simulated mean, mean bias (MB), normalized mean bias (NMB), root
mean square error (RMSE), and correlation coefficient (R) between observation and
simulation at different stations are shown in Table S1 in the SI. The calculations
indicate that T2 is well simulated with MB of −4.25, −3.62, 0.03, and 0.07, and R of
0.69, 0.87, 0.94, and 0.96 at Nam Co, QOMS, Kanpur, and Lahore, respectively. RH2
and U10 are less well simulated, especially at Nam Co and QOMS stations, where the



altitude is very high and the terrain is fairly complex. To be exact, RH2 with MB of
29.68 and −25.82 and R of 0.51 and 0.52 are obtained at Nam Co and QOMS,
respectively. And U10 with MB of 0.91 and 5.66 and R of 0.30 and 0.55 are detected
at two stations accordingly. However, at Kanpur and Lahore stations, RH2 and U10
from reanalysis and simulation are in high consistence, with MBs of −12.56 and 0.85,
−13.00 and 0.96, Rs of 0.80 and 0.45, 0.78 and 0.22, respectively. As a whole, U10 is
on the average overestimated, and has a greater range of about 3.43–8.56 m/s compared
to reanalyzed values with a range of about 2.47–3.39. Hence, the simulations are bias
at least in part since the model grid represents a regional average at $15 \times 15 \ km^2$ in a
domain of great topographic complexity, and the values derived from reanalysis
represent a regional average of a relatively higher resolution. In addition, the accuracy
of the gridded observational data of ERA-interim is correlated with the restriction of
the observations assimilated into the reanalysis and with the different assimilation
methods (Chung et al., 2013). Overall, we conclude that the WRF-Chem model exhibits
acceptable performance in simulating temporal variations in meteorological elements.

The spatial distributions of T2, RH2, and wind field at 500 hPa from simulation

and reanalysis over the domain are presented in Figure S3 in the SI. Spatially, both
simulation and reanalysis show similar spatial patterns for each of the above mentioned
meteorological field. Exactly, surface air temperature with high values mainly appears
over regions surrounding the TP, especially obvious over South Asia where surface air
temperature exceeds 30 °C. Previous studies indicated that surface air temperature over
the Indian subcontinent is the highest during the pre-monsoon season because the





Himalayas block the frigid katabatic winds flowing down from Central Asia during this
period (Ji et al., 2011). In contrast, surface air temperature with low values mainly
occurs over the TP. Different from T2, RH2 with high values primarily appears over
the TP, the Bay of Bengal, and central and eastern China but with low values in the rest
area of the domain. Particularly, RH2 in the southeastern TP is apparently higher than
that of the inland TP, because the southeastern TP is proximity to the moisture sourced
from the Bay of Bengal. Moreover, the reason why the spatial distribution of T2 is
antiphase with that of RH2 is that the decrease in T2 can lead to a decrease in the
saturation pressure of water vapor and an increase in RH2 at the surface (Gao et al.,
2015b). In respect to 500 hPa wind field, both simulation and observation show that the
westerly winds prevail over the entire region. Due to the high topography of the TP,
such westerly winds are divided into two branches at appropriately 75 °E. One branch
flow eastward, and the other branch is forced up by the high plateau and subsequently
shifts to northwesterly wind. Therefore, the WRF-Chem model also effectively
simulates the spatial distributions of T2, RH2, and 500 hPa wind field. Overall, this
simulation configuration captures the meteorological fields well, which is critical to
assure simulation accuracy of air pollutant concentrations.
*3.2.2 Validation of model performance on AOD and BC*

To validate the model performance on simulating spatiotemporal variations in

aerosols, AOD data from AERONET are compared to those from simulation firstly.
Figure S4 shows the temporal variations in simulated and observed daily mean AOD at
Nam Co, QOMS, and Pokhara stations for the period from April 20 to May 10, 2016.



As a whole, the WRF-Chem model reasonably reproduces the temporal variations in
AOD at each of the above stations, with relatively larger bias at Nam Co and Pokhara
stations and minor bias at QOMS station. The specific statistics for N, observed mean,
simulated mean, MB, NMB, RMSE, and R between observed and simulated AOD at
different stations are shown in Table S2 in the SI. The results indicate that MB with
values of −0.13, −0.01, and −0.57, R with values of 0.58, 0.42, and 0.56 are obtained at
Nam Co, QOMS, and Pokhara, respectively. Moreover, the AOD from observation is
significantly correlated with that from simulation at Nam Co and Pokhara stations, with
correlation coefficient passing the 95% confidence level. In addition, we note that the
AOD from simulation is on the average lower than that from observation, which may
be due to the assumed spherical aerosol particles in the model simulation. Actually the
optical properties of particles are more sensitive to non-spherical morphology than
primary spherical structure (China et al., 2015;He et al., 2015). On the whole, the model
effectively reproduces the observed temporal variation in AOD.
Spatially, AOD from both simulation and observation shows distinct spatial
distribution characteristics (Figure S5). The simulated and observed high AOD values
appear over northern South Asia, the Bay of Bengal, Southeast Asia, and the Sichuan
Basin, but low AOD values occur over the TP. This is because South Asia, Southeast
Asia, and Sichuan Basin are heavily industrialized and densely populated regions
compared to the TP (Bran and Srivastava, 2017). In the Taklimakan Desert, AOD
monitored by satellite is much higher than that obtained from simulation, which might
be likely due to the uncertainty of the emission inventory. The comparison between



simulation from WRF-Chem and observation from MODIS and AERONET shows that
the WRF-Chem model captures the overall spatio-temporal characteristics of AOD over
the domain.

To verify the capability of this framework of WRF-Chem on simulating BC

concentration, we present the temporal variation in simulated and reanalyzed hourly
BC concentration at Nam Co, QOMS, Lhasa, NCO-P, Laohugou, and Kanpur stations
during the period from April 20 to May 10, as shown in Figure 4. It is found that the
WRF-Chem model overall reproduces the temporal variation in reanalyzed BC
concentrations at different stations. The specific statistics for N, observed mean,
simulated mean, MB, NMB, RMSE, and R between reanalyzed and simulated BC
concentrations at different stations are shown in Table S2 in the SI. As can be seen, MB
with values of −0.07, 0.14, −0.02, −0.02, 0.02, and 0.72, and R with values of 0.67,
0.43, 0.47, 0.50, 0.25, and 0.64 are obtained at Nam Co, QOMS, Lhasa, NCO-P,
Laohugou, and Kanpur, respectively. The BC concentrations from simulation are
strongly correlated with those from reanalysis at each of the stations, with correlation
coefficient exceeding the 99% confidence level. Hence, the WRF-Chem model exhibits
a better performance in simulating BC concentrations.

Figure S6 presents the spatial distributions of simulated and reanalyzed BC

concentrations over the domain. It can be found that BC concentrations from both
simulation and reanalysis display distinct spatial variability, with low concentrations
over the TP and high concentrations over north of South Asia, Southeast Asia, and
Sichuan Basin. As one of the most pristine regions on the earth, the TP has a small



population density and a low degree of industrialization. Nonetheless, regions like north
of South Asia, Southeast Asia, and Sichuan Basin with low elevation surrounding the
TP have dense population and developed industrialization (Li et al., 2016a;Qin and Xie,
2012;Li et al., 2016b), emitting large amounts of BC into the atmosphere. Therefore,
the WRF-Chem model can capture the main temporal and spatial features of BC
concentrations over the TP and adjacent regions.

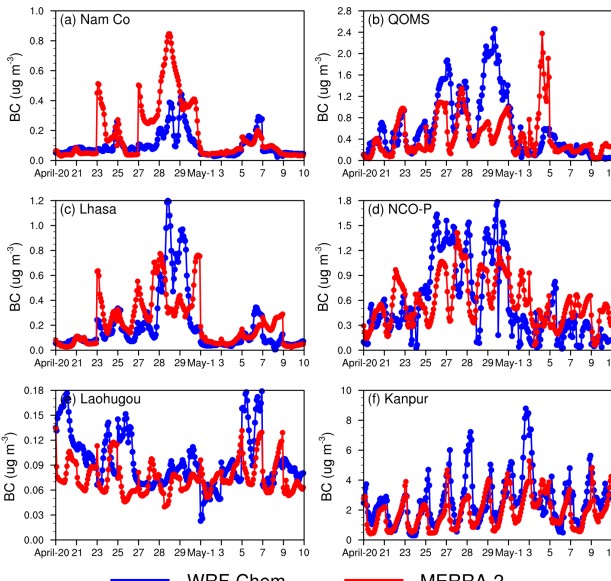


Figure 4 Temporal variations in simulated and reanalyzed hourly BC concentrations at
Nam Co (a), QOMS (b), Lhasa (c), NCO-P (d), Laohugou (e), and Kanpur (f) stations
for the period from April 20 to May 10, 2016.

*3.3 Trans-boundary transport flux of BC*
The foregoing analysis has validated the model framework used in this study and
the results are basically satisfactory. BC, as the major component of light absorbing



particles, exerts a significant impact on climatic and cryospheric changes over the TP
due to its strong light absorption and important effect on snow and ice albedo (Kang et
al., 2010;Kang et al., 2019b;Yang et al., 2018). In this section, the trans-boundary
transport flux of BC during this severe aerosol pollution event is investigated.
According to a previous study, the transport flux can be calculated by projecting the
wind field perpendicularly to the cross line and then multiplying the BC mass
concentration along the cross line (Zhang et al., 2020). More specifically, the transport
flux is calculated as follows:
$$\text{TF} = C \cdot (u \cdot \sin\ \alpha + v \cdot \sin\ \beta), \tag{1}$$
where α is the angle between the east–west wind component and the cross line, β
is the angle between the south–north wind component and the cross line, and C is the
BC mass concentration at the grid along the cross line. The flux is estimated at each
model level. Positive values represent the transport towards the TP, while negative
values represent the transport away from the TP. Figure 5 presents the longitude-height
cross section of BC transport flux along the cross line (shown as the blue dashed line
in Figure 2) from the simulation with aerosol-meteorology feedback at 15:00 and 03:00
BJT averaged for the period from April 27 to May 4 (the most polluted period) to
represent daytime and nighttime transport, respectively. Notably, BC is imported into
the TP in the central and western Himalayas (to the west of ~94 °E) during the day and
night, especially obvious at the height of below 7 km, although the transport flux during
the nighttime is much larger than that during the daytime. In the eastern Himalayas
(from 94–98 °E), BC is imported into the TP during the day but exported slightly from



the TP during the night. To the east of ~98 °E, the BC is transported away from the TP
during the day and night due to the prevailed westerly winds. The transport across the
western Himalayas is controlled by the large-scale westerly, while the transport across
the central and eastern Himalayas is primarily dominated by a local southerly (Zhang
et al., 2020). Therefore, the difference in BC transport flux between the western and
eastern Himalayas is attributed to the influence of a large-scale westerly that is weak
over the eastern Himalayas. The stronger diurnal variation of local southerly (towards
the TP in the daytime to away from the TP in the nighttime) compared to that of a
westerly near the surface leads to the large difference in diurnal variation of the
transport between the western and eastern Himalayas. In addition, the largest BC
transport flux along the cross line occurs at deeper mountain valley channels (Figure 5).
Zhang et al. (2020) investigated the impact of topography on BC transport to the
southern TP during the pre-monsoon season and found that the BC transport across the
Himalayas could overcome the majority of mountain ridges, but the valley transport is
more efficient, which is consistent with the results obtained in this study.

As the largest BC transport flux occurs at deeper mountain valleys, the two deepest

mountain valley channels along the cross line shown as the blue solid line in Figure 2
are selected to demonstrate the transport flux of BC mass across mountain valleys
during this severe pollution event. The first valley (referred to as valley-1 hereon) is
located in the southwestern TP, while the second valley (referred to as valley-2 hereon)
is located in the southeastern TP. It is seen that, at valley-1, the overall positive values
near the surface indicate that BC is imported into the TP during the daytime and



nighttime, though the transport flux at night is much larger than that during the daytime
(Figure 5). Averaged BC concentration and transport flux at 03:00 and 15:00 BJT during
the period from April 27 to May 4, 2016 across valley-1 from the simulation with
aerosol-meteorology feedback also shows that the BC transport flux is much higher
during the night than that during the daytime (Figure 6a–b). By checking the surface
BC concentration from simulation with aerosol-meteorology feedback, it is found that
the surface BC concentration at valley-1 during the night is much higher than that
during the daytime (Table 1). Moreover, the vertically integrated BC concentration over
the study area exhibits higher cross-Himalayan transport of BC during the night than
that during the daytime (Figure S7), which further provides evidence for the higher BC
transport flux during the night shown in Figure 6b. At valley-2, the near-surface positive
flux values during the daytime and negative values during the night denote that BC is
imported into the TP during the daytime but exported slightly from the TP at night
(Figure 5). To analyze this distinct diurnal variation in BC transport flux, we present
the latitude–height cross section of BC transport flux and its concentration at 03:00 and
15:00 BJT averaged for the period from April 27 to May 4, 2016 across valley-2 from
the simulation with aerosol-meteorology feedback as shown in Figure 6c–d. Notably,
the deeper PBLH and the strong turbulent mixing during the daytime over northern
India allows BC to be mixed at a higher altitude (Figure 6c). Subsequently, the local
southerlies boost the BC transporting across the eastern Himalayas towards the TP.
Nevertheless, during the night, the meridional wind is dominated by a northerly over
the eastern Himalayan region (Figure 6d), which indicates that the cross-Himalayan



transport is away from the TP.

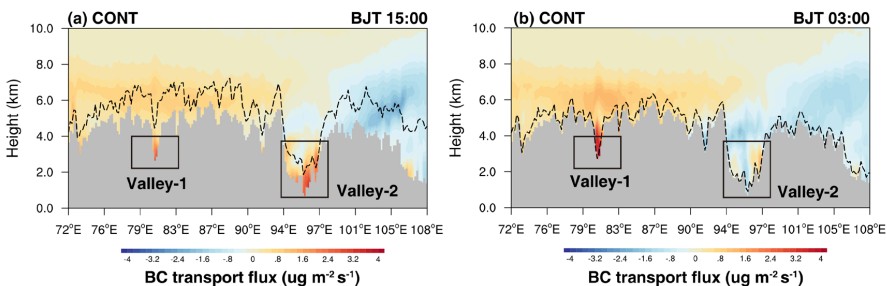

Figure 5 Longitude–height cross section of BC transport flux along the cross line
(shown as the blue dashed line in Figure 2) at 15:00 and 03:00 BJT averaged for the
period from April 27 to May 4, 2016 from simulation with aerosol-meteorology
feedback. The PBLH along the cross section is shown here as the black dashed line.

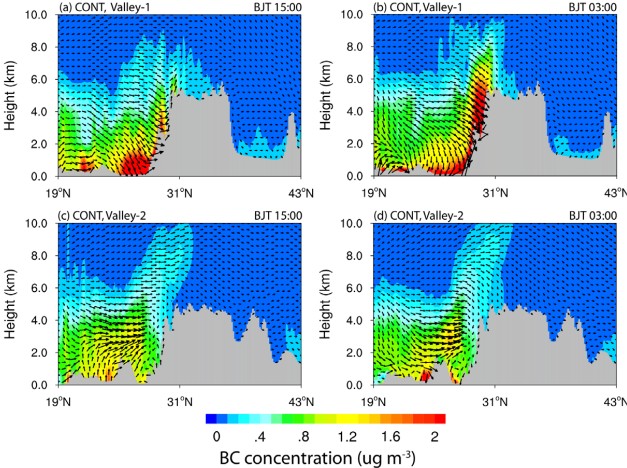

Figure 6 Latitude–height cross section of BC transport flux (vector) across the mountain
valley-1 (a, b) and valley-2 (c, d) at 15:00 and 03:00 BJT averaged for the period from
April 27 to May 4, 2016 from the simulation with aerosol-meteorology feedback.
Contour represents the BC concentration.




Table 1. Surface BC concentration at two typical mountain valley channels at 15:00 and
03:00 BJT averaged for the period from April 27 to May 4, 2016 from simulation with
aerosol-meteorology feedback.

| Near-surface BC concentration | 15:00 | | 03:00 | |
|---|---|---|---|---|
| | Valley-1 | Valley-2 | Valley-1 | Valley-2 |
| CONT | 1.27 | 0.75 | 2.55 | 0.46 |


To further demonstrate the overall inflow flux across the Himalayas, the vertically
integrated BC mass flux along the longitudinal cross section as shown in Figure 5 from
simulation with aerosol-meteorology feedback is shown in Figure 7. The total mass flux
is calculated by integrating the right-hand term of Eq. (1) as follows:
$$ITF = \int_{z=z_{sfc}}^{z=z_{top}} \delta z \cdot C \cdot (\mu \cdot sin\alpha + v \cdot sin\beta), \qquad (2)$$

where δz is the thickness of each vertical model level. Similarly, positive flux
values represent the transport towards the TP, while negative values represent the
transport away from the TP. It has been found that the positive values primarily exist in
the central and western Himalayas (to the west of 92 °E), while negative values mainly
exist to the east of 92 °E (Figure 7). Furthermore, the vertically integrated transport flux
of BC along the cross line is strongly correlated with the longitudinal degree of the
cross line, with the correlation coefficient up to −0.89, passing the 99% confidence level.
This indicates that the vertically integrated transport flux of BC along the cross line
decreases from west to east. In particular, from 80 to 86 °E along the cross line, the

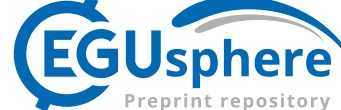

correlation coefficient between the terrain height and the vertically integrated transport
flux of BC is −0.87, exceeding the 99% confidence level, suggesting that the lower the
valleys are, the higher the vertically integrated transport flux transported across the
Himalayas can be. Particularly, the largest vertically integrated transport flux about 20.8
mg m$^{-2}$ s$^{-1}$occurs at valley-1. However, to the eastern Himalayas (to the east of ~92 °E),
the BC is overall exported away from the TP and the vertically integrated transport flux
with the largest value near to zero occurs at valley-2.

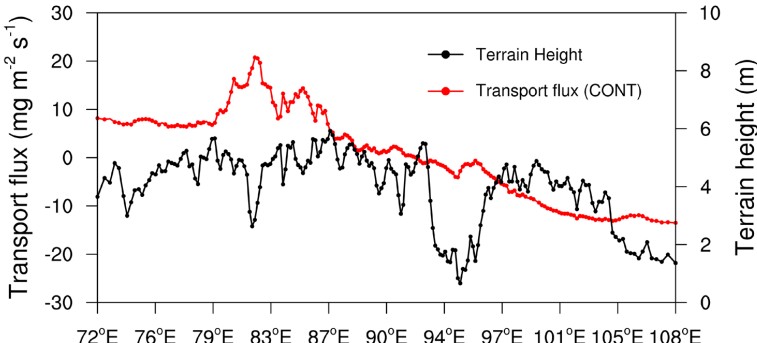


Figure 7 Longitudinal distribution of vertically integrated BC mass flux (red line) along
the cross section in Figure 2 from simulation with aerosol-meteorology feedback. The
black line represents the terrain height.

*3.4 Feedback between aerosol and meteorology*
Generally, the severe aerosol pollution event is accompanied by complex feedback
between aerosol and meteorology. The analysis above confirms that BC in northern
South Asia can be transported to the TP via the cross-Himalayan transport during this
severe pollution event. Moreover, compared to the eastern Himalayas, the western
Himalayas contributes more BC to the TP and BC from the cross-Himalayan transport





mainly concentrated in the southern part of the TP. Therefore, in this section, the
feedback between aerosol and meteorology over the western Indo-Gangetic Plain
(referred to IGP hereon) and southern TP during this severe pollution event is analyzed.
To illustrate the aerosol radiative forcing (ARF) and its impacts on T2, RH2,
surface energy, atmospheric stability, wind, and PBLH over the southern TP and IGP
regions, the time series of aerosol-induced daily and diurnal changes in meteorological
variables (T2, RH2) and surface energy budget (latent heat (LH), sensible heat (SH),
shortwave (SW) radiation, longwave (LW) radiation, and net energy flux
(LH+LW+SH+SW)) averaged for the southern TP and IGP regions, which is calculated
by subtracting the model results of SEN from those of CONT, is shown in Figure 8. It
should be noted that the diurnal change is calculated for the period from April 27 to
May 4, the most polluted period. As can be seen, daily variation of aerosol-induced
area-averaged surface air temperature ranged from −0.1 to 0.1 °C in the southern TP,
with a discernable decrease of 0.1 °C appearing on May 2, May 4, and May 6–7 (Figure
8a), and from −1.7 to 1.2 °C in the IGP (Figure 8c) during the period from April 20 to
May 10. During the most polluted period from April 27 to May 4, aerosol-induced
surface air temperature ranged from −0.1 to 0.1 °C in the southern TP (Figure 8a), with
a decrease of 0.1 °C on May 2 and May 4, and decreased by 0.5–1.7 °C in the IGP
(Figure 8c). Daily variation of aerosol-induced area-averaged RH2 displayed a slight
change with values ranging from −1.6% to 2.3% over the southern TP (Figure 8a) and
exhibited a greater range of about −10.9%–13.7% in the IGP (Figure 8c) during the
period from April 20 to May 10. From April 27 to May 4 with high aerosol



concentrations, area-averaged RH2 increased by 0–2.3% and by 0.8%–13.7% in the
southern TP and IGP, respectively (Figure 8a, c). For the diurnal change depicted in
Figure 8b and Figure 8d, during 09:00–20:00 BJT in the daytime, aerosol-induced area-
averaged surface air temperature had a range of about −0.1–0.1 °C in the southern TP
and decreased by 0.9–2.3 °C in the IGP. At night, surface air temperature increased by
0.1°C during 00:00–05:00 BJT in the southern TP (Figure 8b), and decreased by 0–
1.0 °C during 21:00–08:00 BJT in the IGP (Figure 8d). The diurnal change in aerosol-
induced RH2 decreased by 0.1%–0.4% during 14:00–17:00 BJT in the southern TP and
increased during the rest time of the day, with the maximum increase of 1.3% occurring
during 09:00–10:00 BJT (Figure 8b). In the IGP, aerosol-induced RH2 increased by
3.3%–7.1% during 09:00–20:00 BJT in the daytime and increased by 2.6%–4.5%
during 21:00–08:00 in the nighttime (Figure 8d). Therefore, aerosol-induced changes
in T2 and RH2 primarily occur in the daytime.

Generally, surface energy with positive values indicates more energy flux toward

the surface, and vice versa. Figure 8e–8h present the aerosol-induced surface energy
changes over the southern TP and IGP, it is found that the SW radiation flux at the
surface decreased by 0–13.7 $Wm^{-2}$ in the southern TP (Figure 8e) and by 44.5–75.3
$Wm^{-2}$ in the IGP (Figure 8g) during the period from April 27 to May 4 due to aerosol
scattering and absorption of solar radiation. In contrast, LW radiation flux at the surface
of southern TP and IGP increased due to the positive radiative forcing of aerosol in the
atmosphere, with an increase of 4.1–4.6 $Wm^{-2}$ from April 30 to May 1 in the southern
TP (Figure 8e) and a large increase of 12.3–23.4 $Wm^{-2}$ from April 27 to May 4 in the



IGP (Figure 8g). Because of the cooling effect of aerosols, the LH and SH fluxes from
the surface to the atmosphere in both regions decrease. Particularly, in the southern TP,
the LH and SH fluxes with the maximum decreases of 2.0 $\mathrm{Wm^{-2}}$ and 7.8 $\mathrm{Wm^{-2}}$ occurred
on April 30 and April 29, respectively (Figure 8e). In the IGP, the LH and SH fluxes
decreased by 4.1–7.4 $\mathrm{Wm^{-2}}$ and by 26.6–43.6 $\mathrm{Wm^{-2}}$ from April 27 to May 4,
respectively (Figure 8g). The net energy flux at the surface decreased by 1.6–18.8 W
$\mathrm{m^{-2}}$ in the southern TP (Figure 8e) and by 62.7–104.3 $\mathrm{Wm^{-2}}$ in the IGP (Figure 8g)
from April 27 to May 4. Therefore, the energy arriving at the surface in the southern TP
and IGP decrease during this severe aerosol pollution event. For the diurnal change of
surface energy, during 09:00–20:00 BJT in the southern TP, the SW, LH, SH, and net
energy fluxes decreased by 7.8–19.9 $\mathrm{Wm^{-2}}$, 0–2.4 $\mathrm{Wm^{-2}}$, 4.7–12.2 $\mathrm{Wm^{-2}}$, and 13.4–29
$\mathrm{Wm^{-2}}$, respectively, while the LW flux increased by 0.3–2.9 $\mathrm{Wm^{-2}}$ (Figure 8f). During
09:00–20:00 BJT in the IGP region, the SW, LH, SH, and net energy fluxes decreased
by 46.1–144.2 $\mathrm{Wm^{-2}}$, 7.5–16.6 $\mathrm{Wm^{-2}}$, 9.6–100.1 $\mathrm{Wm^{-2}}$, and 54.7–221.5 $\mathrm{Wm^{-2}}$,
whereas the LW flux increased by 8.5–30.5 $\mathrm{Wm^{-2}}$ (Figure 8h). Therefore, changes in
surface energy mainly occur during the daytime.



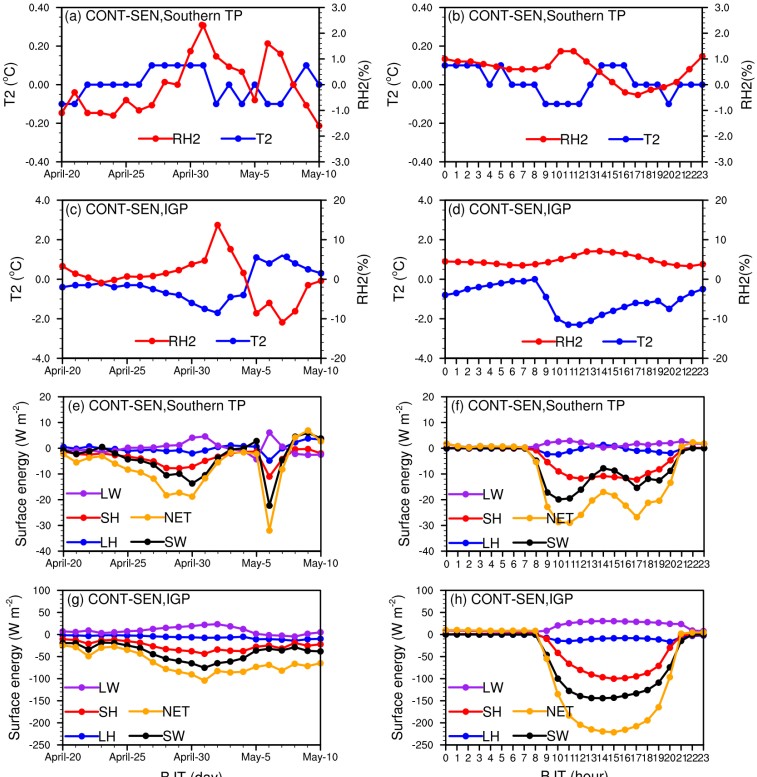

Figure 8 Time series of aerosol-induced daily changes in (a, c) meteorological variables (T2 (°C), RH2 (%)) and (e, g) surface energy budget (SH, LH, LW radiation, SW radiation, and net energy flux, Wm$^{-2}$) averaged for the (a, e) southern TP and (c, g) IGP during the period from April 20 to May 10. Time series of aerosol-induced diurnal changes in (b, d) meteorological variables and (f, h) surface energy budget averaged for the period from April 27 to May 4, 2016 for the (b, f) southern TP and (d, h) IGP. LH is latent heat, LW is long-wave radiation, SH is sensible heat, SW is shortwave radiation, and NET is the sum of the total energy fluxes.

Figure 9 shows the spatial distribution of aerosol-induced changes in T2 and RH2



and aerosol radiative forcing (ARF) at the bottom of atmosphere as well as in the
atmosphere, calculated by subtracting the model results of SEN from those of CONT
averaged during 09:00–20:00 BJT from April 27 to May 4. As seen in Figure 9a, the
aerosol-induced surface air temperature decreased over most parts of the study area
except for the TP, where surface air temperature increased, especially obvious in the
northern TP, with surface air temperature increasing by up to 1.0 °C. Because the
northern TP is proximity to the Taklimakan Desert, where desert dust type aerosols
dominate. The strong light absorption of dust aerosols from the Taklimakan Desert
increases surface air temperature over the northern TP. Surface air temperature decrease
indicates that solar shortwave radiation reaching the surface decreases due to the
dimming effect of aerosols. In particular, the largest surface air temperature decrease
due to aerosols occurred in South Asia. Because South Asia has large amounts of
aerosols due to rapid economic growth, industrialization, and unplanned urbanization
compared to other regions (Shi et al., 2020). The spatial distribution of aerosol-induced
changes in RH2 is opposite to that of T2, with increased RH2 appearing in most parts
of the study area and decreased RH2 on the TP. The largest increase in RH2 occurred
in northern South Asia, since the aerosol-induced changes in water vapor mixing ratio
is very small, and the decrease in temperature can lead to a decrease in the saturation
pressure of water vapor and an increase in RH2 at the surface, which is beneficial for
the hygroscopic growth of aerosols (Gao et al., 2015b). By analyzing the time–altitude
distribution of the diurnal cycle of aerosol impacts on temperature and relative humidity
(RH) averaged for the southern TP and IGP during the period from April 27 to May 4,



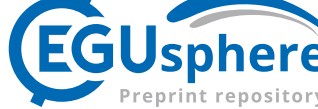

it is found that, consistent with aerosol-induced changes in T2 and RH2, the changes in
temperature and RH mainly occurred during daytime as well (Figure S8). Specifically,
in the southern TP, the maximum increase in temperature with value up to 0.15 °C
occurred in the middle troposphere (Figure S8a). However, in the IGP, the temperature
decreased near the surface with a maximum drop of more than 0.3 °C and increased in
the middle troposphere with a maximum increase of more than 0.3 °C (Figure S8c).
There is no doubt that such a temperature change increases the stability of the
atmosphere over both regions. Note that the temperature increase in the middle
troposphere over the southern TP is more significant than that in the IGP, which is
possibly correlated with the thermal pump role of the TP (Li and Yanai, 1996;Meehl,
1994;Yanai et al., 1992). The time–altitude distribution of the diurnal cycle of RH is
opposite to that of temperature, with RH increasing near the surface and decreasing in
the middle troposphere (Figure S8b and Figure S8d). To be exact, RH decreased by 1.6%
in the middle troposphere over the southern TP (Figure S8b), while in the IGP, RH
increased near the surface with value greater than 3% and decreased by more than 3%
in the middle troposphere (Figure S8d).
The ARF at the bottom of the atmosphere is negative over most parts of the study
area except for the northern TP, where ARF is positive, with ARF up to 20 W m$^{-2}$
(Figure 9c). The observed largest negative ARF occurred in northern South Asia and
the Bay of Bengal, with values in a range of about −40−−120 Wm$^{-2}$ (Figure 9c).
Contrary to the spatial distribution of ARF at the bottom of the atmosphere, the ARF in
the atmosphere is positive over most parts of the study area, with the largest ARF up to





110 Wm$^{-2}$ in South Asia (Figure 9d). Thus, affected by aerosols, the atmospheric
stratification over the study area is expected to be more stable.

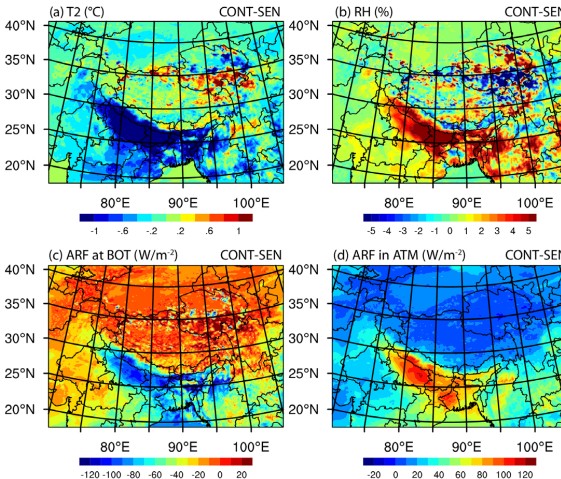


Figure 9 Spatial distribution of aerosol-induced changes in T2 (°C) (a) and RH2 (%)
(b), and aerosol radiative forcing (ARF, Wm$^{-2}$) at the bottom of (BOT) (c) and in the
atmosphere (ATM) (d) averaged for 09:00–20:00 BJT during the period from April 27
to May 4, 2016.

From the above analysis, aerosol-induced changes in meteorological conditions

and ARF have significant impact on atmospheric stability. The profile of equivalent
potential temperature (EPT) can be used to characterize the stability of the atmosphere.
Figure S9 in the SI shows the aerosol-induced changes in EPT profiles at 02:00, 08:00,
14:00, and 20:00 BJT, averaged during the period from April 27 to May 4 in the
southern TP and IGP. It can be seen that aerosol impact on EPT over the southern TP
with values in a range of about 0.08–0.24 K is overall smaller than that in the IGP with
values in a range of about −1.6–3.2 K due to low aerosol concentrations in the southern





TP and high aerosol concentrations in the IGP. Specifically, at 08:00, 14:00 and 20:00
BJT, the EPT in the southern TP decreased with height in the lower and middle
troposphere, and increased with height above the middle troposphere (Figure S9a).
However, in the IGP, at 02:00 and 20:00 BJT, the EPT decreased with height below
550 hPa and increased with height above 550 hPa, while at 14:00 BJT, the EPT
decreased with height below 650 hPa and increased with height between 650 and 550
hPa (Figure S9b). Therefore, an obvious temperature inversion was observed in the
troposphere over the southern TP and IGP during the severe aerosol pollution event.

Under a more stable atmosphere, the diurnal variation of surface BC

concentrations from control experiment with aerosol-meteorology feedback and
aerosol-induced changes in U10 and planetary boundary layer height (PBLH) over the
southern TP and IGP are presented in Figure 10. Overall, BC concentrations over the
southern TP and IGP were high during 20:00–08:00 BJT in the nighttime and low
during 08:00–20:00 BJT in the daytime. Over the southern TP, the aerosol-induced
PBLH decreased with value up to 55 m during 09:00–14:00 BJT and by 70 m during
18:00–20:00 BJT, while at other times of the day, no obvious change in PBLH was
observed (Figure 10a). In the IGP, the aerosol-induced PBLH decreased during 10:00–
20:00 BJT in the daytime, with the largest decrease of 1700 m occurring at 20:00 BJT
(Figure 10b). Lower PBLH constrains the pollutants to diffuse in the vertical direction
and is conducive to the accumulation of pollutants near the ground. Aerosol-induced
U10 increased in the southern TP, with the maximum increase of 0.2 m/s appearing at
19:00 BJT (Figure 10a). In the IGP, the aerosol-induced U10 decreased, with the largest





decrease of 0.7 m/s appearing at 20:00 BJT (Figure 10b). Therefore, aerosol induces
significant changes in meteorological conditions in the southern TP and IGP during this
severe aerosol pollution event.

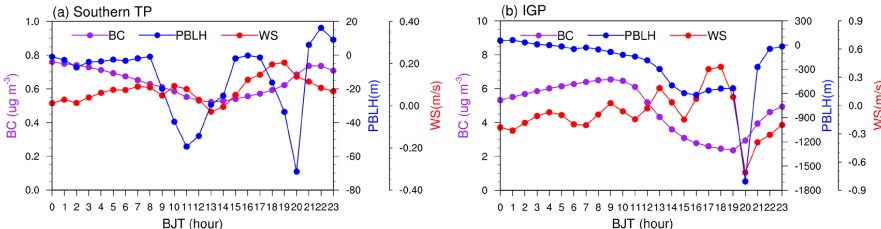


Figure 10 Diurnal variation of surface BC concentrations ($\mu$g m$^{-3}$) from control
experiment with aerosol-meteorology feedback (purple solid line) and aerosol-induced
diurnal changes in 10-m wind speed (red solid line, WS, m s$^{-1}$) and PBLH (blue solid
line, m) averaged for (a) the southern TP and (b) IGP during the period from April 27
to May 4, 2016.

Aerosol-induced changes in meteorological conditions have important effect on

surface BC concentration, which eventually exerts potential influence on aerosol
pollution as well as weather and climate (Menon et al., 2002). Figure 11a and b show
the hourly surface BC concentration from sensitive experiment without aerosol-
meteorology feedback and the impact of aerosol-induced changes in meteorological
variables on hourly surface BC concentration averaged over the southern TP and IGP
during the period from April 20 to May 10. The corresponding diurnal cycle during the
most polluted period from April 27 to May 4 are shown in Figure 11c and d. The change
in percentage is calculated by comparing with the surface BC concentration from the





experiment without aerosol-meteorology feedback. It can be seen that aerosol-induced
changes in meteorological conditions lead to a decrease of surface BC concentration
with value up to 0.16 μg/m$^3$ (50%) in the southern TP. However, in the IGP, aerosol-
induced changes in meteorological conditions result in an increase in surface BC
concentration with value up to 2.2 μg/m$^3$ (75%). Moreover, the higher the surface BC
concentration is, the greater the variation in the surface BC concentration induced by
meteorological conditions is. It should be noted that the time when the maximum
decrease or increase in surface BC concentration occurs is not the time when the
maximum surface BC concentration occurs. The diurnal changes in surface BC
concentration during the most polluted period from April 27 to May 4 over the southern
TP indicate that surface BC concentration is high during 20:00–07:00 BJT in the
nighttime and low during 08:00–19:00 BJT in the daytime, with the lowest
concentration of 0.6 μg/m$^3$ observed at 12:00 BJT. The changes in meteorological
conditions lead to a reduction of surface BC concentration, with the reduction primarily
occurring during 11:00–19:00 BJT in the daytime and the largest reduction of 0.06
μg/m$^3$ (12%) appearing at 15:00 BJT. The corresponding diurnal changes in the IGP
reveal that surface BC concentration is high during 23:00–12:00 BJT and low during
13:00–22:00 BJT. The changes in meteorological conditions result in an increase in
surface BC concentration, with the relatively larger increase of about 0.7–1.1 μg/m$^3$
occurring during 20:00–13:00 BJT. Therefore, the changes in meteorological
conditions enhance the diurnal variation of surface BC concentration by decreasing the
surface BC concentration in the southern TP and increasing the surface BC



concentration in the IGP.

Figure S10 in the SI shows the impact of aerosol-induced changes in

meteorological conditions on the spatial distribution of surface BC concentration
averaged during 09:00–20:00 BJT, April 27–May 4. Consistent with Figure 11, the
maximum increase in surface BC concentration induced by changes in meteorological
conditions is in the IGP (northwestern South Asia), with values greater than 1 μg/m³.
The corresponding surface BC concentration change in percentage terms is also higher
in northwestern South Asia with value up to 30%, and is lower in the southeastern TP
with value below 30% (Figure S10b). Taken together, aerosols result in significant
changes in meteorological conditions in the Southern TP and IGP, with an obvious
decrease in PBLH and U10 along with a more stable atmosphere in the IGP, and a
decrease in PBLH but an increase in U10 accompanied by a stable atmosphere in the
Southern TP. In addition, aerosol-induced changes in meteorological conditions have
substantial influence on aerosol concentrations. For instance, in the IGP, changes in
meteorological conditions induced by aerosols are conducive to the accumulation of air
pollutants, thereby contributing to the formation of severe aerosol pollution event.
Consequently, a mechanism of positive feedback exists between aerosol concentration
and aerosol-induced meteorological conditions in the IGP. However, as one of the
major source regions of air pollutants over the TP, aerosol-induced changes in
meteorological conditions in the IGP are not favorable for air pollutants transporting to
the southern TP.



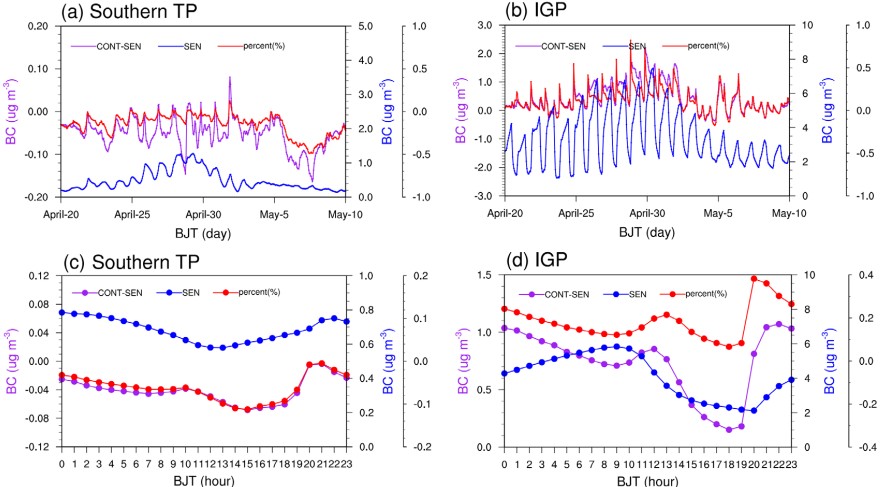


Figure 11 Time series of hourly surface BC concentration from SEN (blue solid line)
averaged for the southern TP (a) and IGP (b) during the period from April 20 to May
10, 2016 and the corresponding diurnal changes (blue solid line) averaged for the
southern TP (c) and IGP (d) during the period from April 27 to May 4, 2016. The purple
solid line denotes the change value of surface BC concentration induced by
meteorological conditions. The red solid line indicates the corresponding change in
percentage terms compared to the surface BC concentration from the model results of
SEN.

*3.5 Impact of aerosol-meteorology feedback on the trans-boundary transport flux of*
*BC*
As discussed above, during the severe aerosol pollution event, northwestern South
Asia contributes more BC to the TP via cross-Himalayan transport and the largest BC
transport flux occurs at mountain valley in western Himalayas. Moreover, the aerosol-
meteorology feedback has a substantial effect on surface BC concentration over the





southern TP and IGP. Yet what effect the aerosol-meteorology feedback having on the
trans-boundary transport flux of BC remains unclear, which deserves further
investigation. Therefore, this section is aimed at demonstrating the impact of aerosol-
meteorology feedback on the trans-boundary transport flux of BC during the severe
aerosol pollution event. Figure 12 shows the difference in longitude–height cross
section of BC transport flux along the cross line (shown as the blue dashed line in Figure
2) from the simulations with and without aerosol-meteorology feedback at 15:00 and
03:00 BJT averaged for the period from April 27 to May 4, 2016. It can be seen that, in
the central and western Himalayas (75–90 °E), the aerosol-meteorology feedback
during the daytime overall increases the BC transport flux towards the TP at the height
of about 6–7 km but decreases the BC transport flux at the height of below 6 km;
however, in the eastern Himalayas (90–98 °E), the aerosol-meteorology feedback
decreases the BC transport flux exporting from the TP. During the nighttime, the
aerosol-meteorology feedback increases the BC transport flux towards the TP at the
height of about 6–7 km but decreases the BC transport flux at the height of below 6 km
in the central and eastern Himalayas (from 80 °E to 94 °E); however, from 94 °E to
98 °E in the eastern Himalayas, the aerosol-meteorology feedback decreases the BC
transport flux exporting from the TP.

In particular, the impact of aerosol-meteorology feedback on the BC transport flux

at two typical mountain valley channels reveals that, at valley-1, the feedback decreases
the import of BC towards the TP during the daytime and nighttime; while at valley-2,
the feedback decreases the import of BC towards the TP during the daytime and reduces



804 the BC transport flux away from the TP during the nighttime. To better understand the

805 effect of aerosol-meteorology feedback on the BC transport flux at valley-1 and valley-

806 2, we investigated the mean zonal and meridional wind speeds within 3500 m above

807 the ground level at both valleys during the daytime and nighttime from April 27 to May

808 4, 2016 (Table 2) from simulations with and without aerosol-meteorology feedback. It

809 is found that, during the daytime, a westerly and a southerly prevail at valley-1 but an

810 easterly and a southerly prevail at valley-2; during the night, an easterly and a northerly

811 prevail at both valleys in both experiments. Specifically, at valley-1, the differences in

812 zonal and meridional wind speeds between the simulations with and without aerosol-

813 meteorology feedback at 15:00 and 03:00 BJT averaged for the period from April 27 to

814 May 4, 2016 show that, during the daytime, the aerosol-meteorology feedback overall

815 decreases the westerly and southerly wind speeds, resulting in decreased transport flux

816 of BC mass towards the TP; during the night, the aerosol-meteorology feedback leads

817 to increased easterly and northerly wind speeds, which strengthen the BC transport flux

818 away from the TP. The corresponding results at valley-2 indicate that, during the

819 daytime, the aerosol-meteorology feedback increases the easterly wind speed but

820 decreases the southerly wind speed, resulting in decreased transport flux of BC towards

821 the TP; during the night, the aerosol-meteorology feedback increases the easterly wind

822 speed but decreases the northerly wind speed, leading to reduced BC transport flux

823 away from the TP. It is to be emphasized that the results averaged within 3900 m above

824 the ground level (Table S3) are consistent with those averaged within 3500 m.



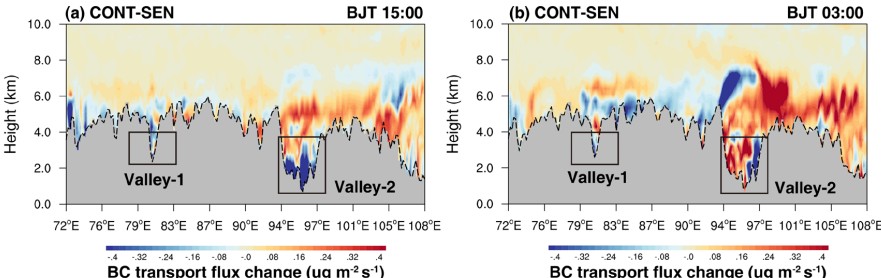

Figure 12 Difference in longitude–height cross section of BC transport flux along the cross line (shown as the blue dashed line in Figure 2) from the simulations with and without aerosol-meteorology feedback at 15:00 and 03:00 BJT averaged for the period from April 27 to May 4, 2016. The difference in PBLH along the cross section is shown here as the black dashed line.

Table 2. The mean zonal and meridional wind speeds at two typical valley channels within 3500 m above the ground level at 15:00 and 03:00 BJT averaged for the period from April 27 to May 4, 2016 between the CONT and SEN experiments. The differences in zonal and meridional wind speeds between the two experiments are also shown. Positive value denotes a westerly or a southerly and negative value denotes an easterly or a northerly.

| 3500 m | | 15:00 | | 03:00 | |
|---|---|---|---|---|---|
| | | Valley-1 | Valley-2 | Valley-1 | Valley-2 |
| CONT | U component | 2.32 | −1.90 | −1.34 | −1.19 |
| | V component | 4.74 | 1.35 | −1.73 | −1.56 |
| SEN | U component | 2.75 | −1.43 | −1.26 | −1.07 |
| | V component | 4.94 | 1.61 | −1.60 | −1.92 |
| CONT−SEN | U component | −0.43 | −0.47 | −0.08 | −0.12 |
| | V component | −0.2 | −0.26 | −0.13 | 0.36 |

Similarly, the impact of aerosol-meteorology feedback on the vertically integrated



trans-boundary transport flux of BC along the cross line shown as the blue dashed line
in Figure 2 reveals that, with the aerosol-meteorology feedback, the integrated transport
flux of BC mass from central and western Himalayas (to the west of 88 °E) to the TP
overall decreases; however, in the eastern Himalayas, the aerosol-meteorology
feedback increases the integrated transport flux of BC towards the TP (Figure 13).
Therefore, aerosol-meteorology feedback exerts a very important effect on trans-
boundary transport flux of BC.
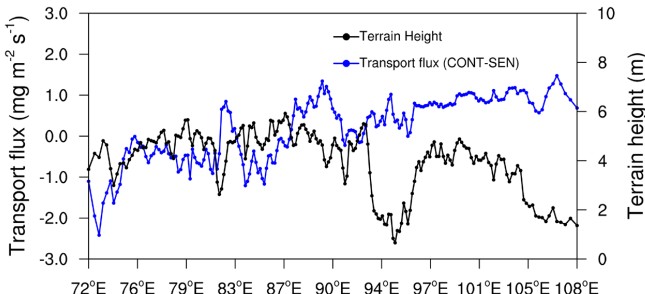
Figure 13 Difference in longitudinal distribution of integrated BC mass flux along the
cross section in Figure 2 from the simulations with and without aerosol-meteorology
feedback. The black line represents the terrain height.

**4 Conclusion**

The worst aerosol pollution episode ever recorded over the TP occurred during the

period from April 20 to May 10, 2016. The observed largest AOD at reference sites of
Nam Co and QOMS are 0.65 and 0.42, respectively. In this paper, the meteorological
causes, BC transport flux, and aerosol-meteorology feedback as well as its effect on BC
transport flux during this severe aerosol pollution event are investigated by using



observational and reanalysis dataset, and numerical simulation from WRF-Chem. By
analyzing the evolution of weather maps at 500 hPa over the study area during this
severe aerosol pollution event, it is found that the plateau vortex plays a critical role in
increasing aerosol concentrations in the inland of the TP. However, in the southern TP,
the increase in aerosol concentration could be attributed to the long-range transport by
southwesterly airflow in front of the trough.

With the acceptable performance of WRF-Chem model on simulating

meteorological conditions and aerosols, we estimated the trans-boundary transport flux
of BC across the Himalayas. The results show that, in the central and western
Himalayas, BC is imported into the TP during the day and night; in the eastern
Himalayas, BC is imported into the TP during the day but exported slightly from the
TP during the night; however, to the east of ∼98 °E, BC is transported away from the
TP during the day and night. The vertically integrated transport flux of BC along the
cross line during the aerosol pollution event decreases from west to east. The largest
vertically integrated transport flux of BC with value up to 20.8 mg m$^{-2}$ s$^{-1}$ occurs at a
mountain valley in southwestern TP.

By designing experiments with or without aerosol-meteorology feedback, the

feedback between aerosol and meteorology over the southern TP and IGP during this
severe aerosol pollution event are investigated. It has been found that during the most
polluted period from April 27 to May 4, aerosols lead to a slight change in surface air
temperature in the southern TP but a significant decrease by 0.5–1.7 °C in the IGP.
Vertically, in the southern TP, the largest temperature increase induced by aerosols



occurs in the middle troposphere; however, in the IGP, aerosol-induced temperature
decreases near the surface but increases in the middle troposphere. Spatially, the ARF
is negative at the bottom of the atmosphere but is positive in the atmosphere over most
parts of the study area. As a result, the atmospheric stratification over the study area is
more stable. Additionally, affected by aerosols, U10 increases in the southern TP, with
the largest increase of 0.2 m/s appearing at 19:00 BJT; while in the IGP, U10 decreases,
with the largest decrease of 0.7 m/s appearing at 20:00 BJT. In respect to PBLH,
aerosols lead to a decrease by 55 m in PBLH during 09:00–14:00 BJT and by 60 m
during 17:00–20:00 BJT in the southern TP; whereas in the IGP, a decrease in PBLH
resulted from aerosols mainly occurs during 10:00–20:00 BJT in the daytime, with the
largest decrease of 1300 m detected at 20:00 BJT. Therefore, aerosols exert an
important effect on meteorological conditions. By contrast, aerosol-induced changes in
meteorological conditions can lead to a decrease of surface BC concentration with value
up to 0.16 μg/m$^3$ (50%) in the southern TP and an increase of surface BC concentration
with value up to 2.2 μg/m$^3$ (75%) in the IGP.

By investigating the impact of aerosol-meteorology feedback on the BC transport

flux, it has been acquired that, with the aerosol-meteorology feedback, the integrated
transport flux of BC mass from central and western Himalayas to the TP overall
decreases; however, in the eastern Himalayas, the aerosol-meteorology feedback
increases the integrated transport flux of BC towards the TP. In particular, the
corresponding results at two typical mountain valley channels in southwestern and
southeastern TP reveal that the aerosol-meteorology feedback decreases the import of





BC towards the TP at mountain valley channel in southwestern TP during the daytime
and nighttime, while at mountain valley channel in the southeastern TP, the feedback
decreases the import of BC towards the TP during the daytime and reduces the BC
transport flux away from the TP during the nighttime.

There are still uncertainties in this study. Because the aerosol feedback derived

from the aerosol radiative effect mainly has large impacts during the daytime. From the
statistical analysis of model performance on aerosols, we find that the model overall
underestimates the AOD. This underestimation may have important effect on aerosol
feedback during the most polluted period. Moreover, the feedback between aerosol and
meteorological conditions also has uncertainty due to the fact that the aerosol direct and
indirect effect is very sensitive to the mixing state between scattering aerosols and
absorbing aerosols. In addition, the BC transport flux quantified by WRF-Chem model
has bias to some extent. However, with the scarce observation over the TP, numerical
model is the best tool for this study. Therefore, emissions with higher resolution and
finer model horizontal resolution could improve model performance, since further in-
depth investigation is deserved.
**Data availability**

All raw data can be provided by the corresponding authors upon request.

**Author contributions**

Shichang Kang and Haipeng Yu provided idea for this paper. Yuling Hu and

Junhua Yang designed the experiments and performed the simulations. Yuling Hu wrote
the code and explained the model results. Pengfei Chen provided the data to verify the



model results. Yuling Hu, Mukesh Rai, Xiufeng Yin, and Xintong Chen prepared,
reviewed and edited the manuscript with contributions from all co-authors.

## Competing interests

The authors declare that they have no conflict of interest.

## Acknowledgments

This study was supported by the National Natural Science Foundation of China
(42205123), the Chinese Academy of Sciences (XDA20040501, QYZDJ-SSW-
DQC039), and the State Key Laboratory of Cryospheric Science (SKLCS-ZZ-2023).
The authors would like to acknowledge the National Centers for Environmental
Prediction (NCEP) and the European Centre for Medium-Range Weather Forecasts
(ECMWF) for providing final analysis data and reanalysis data, respectively.

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
