# Peer review of "Aerosol-meteorology feedback diminishes the trans-boundary"

_EGUsphere, 2023_

## Author Comment (AC1)

**1 Dear editor,**

Thank you for your kind considerations on our manuscript entitled "Aerosol-2 3 meteorology feedback diminishes the trans-boundary transport of black carbon into the Tibetan Plateau" (egusphere-2023-252). We appreciate that you gave us a chance to 4 improve our manuscript to a level suitable for publication in ACP. We also want to 5 6 express our deep thanks to the reviewers of the positive comments. Those comments 7 are all valuable and very helpful for revising and improving our paper. We have studied comments carefully and have made corrections, which we hope meet with approval. 8 The main corrections in the paper and the responds to the reviewer's comments are as 9 following: 10

11

12 Answers to reviewers:

13 Reviewer #1:

The manuscript reported a record-breaking aerosol pollution event in the TP. Cross-14 boundary transport along the south slope the Himalayas from the surrounding regions 15 was clearly the cause, implying glacier melt and ecological environment disturbance 16 for the TP. Due to the extremely scarce availability of field observations, however, 17 cross-boundary transport flux, meteorological pattern delivering aerosol, and the 18 19 aerosol-meteorology feedback have rarely been discussed. The manuscript clearly 20 shows strong aerosol-meteorology feedback on the transport flux of aerosols. The strong feedback on meteorology and aerosol distribution are also discussed in details. I 21 hence recommended this manuscript for publication in ACP. 22

Firstly, we appreciate that you gave us a chance of revision to improve our manuscript to a level suitable for publication in Atmospheric Chemistry and physics. We also want to express our deep thanks to your positive comments. The comments are replied as follows:

27

1. The measurement distinguishes this manuscript and also a previous one, i.e., zhang et al., 2020, written by the key authors from a pure model simulation. However, these major results on transport flux and the aerosol-meteorology feedback are merely model simulations? Whether do the observation here or in other places reflect the pattern of transport flux or the aerosol-meteorology feedback pattern on transport flux?

Response: Thank you for your valuable suggestion. As the reviewer stated, the major
results on transport flux and the aerosol-meteorology feedback are from model
simulations. Because known as the 'Third Pole', the Himalayas and the Tibetan Plateau
(TP) have very limited observational dataset due to harsh environment, limited access
for fieldwork, and the sparsity of fixed instrumental stations.

Through the literature research, it is found that there are studies using observational dataset to reflect the transport flux of aerosols. For example, using ground-based multi-axis differential optical absorption spectroscopy (MAX-DOAS) at the Nancheng site in suburban Beijing on the southwest transport pathway of the Beijing-Tianjin-Hebei (BTH) region, Hu et at. (2022) estimated the vertical profiles of transport fluxes in the southwest-northeast direction. The results showed that the maximum net transport fluxes per unit cross-sectional area, calculated as pollutant

concentration multiply by wind speed, of aerosol extinction coefficient (AEC), NO2, 45 SO2 and HCHO were 0.98 km-1m s-1, 24, 14 and 8.0  $\mu$ g m-2 s-1 from southwest to 46 northeast, which occurred in the 200-300 m, 100-200 m, 500-600 m and 500-600 m 47 layers, respectively, due to much higher pollutant concentrations during southwest 48 transport than during northeast transport in these layers. The average net column 49 transport fluxes were 1200 km-1 m2 s-1, 38, 26 and 15 mg m-1 s-1 from southwest to 50 northeast for AEC, NO2, SO2 and HCHO, respectively, in which the fluxes in the surface 51 layer (0–100 m) accounted for only 2.3%–4.2%. 52

However, in terms of the influence of aerosol-meteorology feedback on transport 53 flux of aerosols, it is found that no matter in regions with abundant observational data 54 or in regions with sparse observational data, the influence of aerosol-meteorology 55 feedback on the transport flux of aerosols was evaluated by means of model simulation, 56 57 because sensitivity tests are involved in such studies. For instance, Huang et al. (2020) suggested that the aerosol-meteorology interaction and feedback enhanced the trans-58 boundary transport of pollutants between the North China Plain and the Yangzi River 59 Delta regions and thus exacerbated the haze levels in these two regions simultaneously, 60 61 which was published in nature geoscience.

62

Although observation-model comparison on BC concentration and AOD have been
 conducted, these results do not fully justify the simulated transport flux pattern? More
 regious comparisons also transport flux among observations/reanalysis and models, or
 intermodel comparison on transport flux, might be helpful?

Response: Thank you for your good advice. The reviewer made a very good point here. 67 68 According to the reviewer's suggestion, we not only validated the model performance on temporal variation in AOD at different stations by comparing the simulated AOD 69 with the ground-based and satellite-based observational AOD (Figure S4), but also 70 71 verified the model performance on the spatial distribution of AOD over the study area by comparing the simulated AOD with the satellite-based and reanalyzed AOD (Figure 72 S5). However, for BC, the comparison between reanalysis and simulation was only 73 conducted because we have very limited in-situ observed BC (the observed BC is only 74 available at the QOMS station). According to the reviewer's suggestion, we further 75 validated the model performance by conducting the inter-comparison among in-situ 76 observation, simulation, and MERRA-2 reanalysis, as shown in Figure A1. The results 77 show that the temporal variation in simulation is very close to that of simulation. 78 79 Moreover, the correlation coefficient between the simulation and observation is 0.867, 80 passing the 99% confidence level. Therefore, the model configuration used in this study 81 presented a reasonable performance on BC. For the spatial distribution of BC over the study area, we compared the simulation with reanalysis from MERRA-2 and simulation 82 from CAM Chem. The results show that the spatial pattern of the WRF-Chem 83 simulated BC is similar to that of the reanalyzed BC (Figure S6); however, the spatial 84 pattern of BC from CAM Chem is not reasonable (Figure A2). 85

Additionally, in terms of the BC transport flux, as the spatial pattern of BC from CAM\_Chem is not reasonable, we can't further verify the transboundary transport flux of BC with CAM\_Chem data. Also, the BC from MERRA-2 has no vertical information,

resulting in the inability to provide vertical profile of BC transport flux. Therefore, thetransport flux of BC was not verified by inter-model comparison.

---

## Author Comment (AC2)

**1 Dear editor,**

Thank you for your kind considerations on our manuscript entitled "Aerosol-2 meteorology feedback diminishes the trans-boundary transport of black carbon into the 3 Tibetan Plateau" (egusphere-2023-252). We appreciate that you gave us a chance to 4 improve our manuscript to a level suitable for publication in ACP. We also want to 5 6 express our deep thanks to the reviewers of the positive comments. Those comments are all valuable and very helpful for revising and improving our paper. We have studied 7 comments carefully and have made corrections, which we hope meet with approval. 8 The main corrections in the paper and the responds to the reviewer's comments are as 9 following: 10

11

12 Answers to reviewers:

13 Reviewer #2:

The interaction between aerosols and meteorology, and its impact on the cross-14 boundary transport flux of BC (black carbon) over the Tibetan Plateau (TP), has 15 received limited attention in previous research. This paper presents a comprehensive 16 investigation of the aerosol-meteorology feedback and its influence on BC transport 17 flux during a period of heavy aerosol pollution. The study utilizes WRF-Chem 18 19 simulation to thoroughly analyze the phenomenon. Additionally, the paper elucidates 20 the meteorological factors that contribute to the occurrence of severe aerosol pollution events over the TP. The concept introduced in this article is characterized by its novelty, 21 and the study's findings hold significant implications for the preservation of the TP's 22 ecological environment. Hence, I recommend that this manuscript be revised and 23 24 considered for publication in ACP. Please find below some specific comments for 25 further improvement:

Firstly, we appreciate that you gave us a chance of revision to improve our manuscript to a level suitable for publication in Atmospheric Chemistry and physics. The comments are replied as follows:

29

The authors validated the model performance on BC and AOD by comparing the
 simulation and observation. Although the comparison results are basically satisfactory,
 the data used to validate the model performance is still simple and I suggest inter-model
 comparison should be considered, which might be more convincing.

Response: Thank you very much for your valuable advice. The reviewer made a very 34 good point here. According to the reviewer's suggestion, we not only validated the 35 model performance on temporal variation in AOD at different stations by comparing 36 37 the simulated AOD with the ground-based and satellite-based observational AOD (Figure S4), but also verified the model performance on the spatial distribution of AOD 38 over the study area by comparing the simulated AOD with the satellite-based and 39 reanalyzed AOD (Figure S5). However, for BC, the comparison between reanalysis and 40 simulation was only conducted because we have very limited in-situ observed BC (the 41 observed BC is only available at the QOMS station). According to the reviewer's 42 suggestion, we further validated the model performance by conducting the inter-43 comparison among in-situ observation, simulation, and MERRA-2 reanalysis, as shown 44

in Figure A1. The results show that the temporal variation in simulation is very close 45 to that of simulation. Moreover, the correlation coefficient between the simulation and 46 observation is 0.867, passing the 99% confidence level. Therefore, the model 47 configuration used in this study presented a reasonable performance on BC. For the 48 49 spatial distribution of BC over the study area, we compared the simulation with 50 reanalysis from MERRA-2 and simulation from CAM Chem. The results show that the spatial pattern of the WRF-Chem simulated BC is similar to that of the reanalyzed BC 51 (Figure S6); however, the spatial pattern of BC from CAM Chem is not reasonable 52 53 (Figure A2).

Additionally, in terms of the BC transport flux, as the spatial pattern of BC from CAM\_Chem is not reasonable, we can't further verify the transboundary transport flux of BC with CAM\_Chem data. Also, the BC from MERRA-2 has no vertical information, resulting in the inability to provide vertical profile of BC transport flux. Therefore, the transport flux of BC was not verified by inter-model comparison.

59

60 Figure S4. Inter-comparison of temporal variations in simulated AOD and ground-61 based as well as satellite-based AOD at (a) Nam Co, (b) QOMS, and (c) Pokhara 62 stations for the period from April 20 to May 10, 2016

62 stations for the period from April 20 to May 10, 2016.

63 64

65

Figure S5. Inter-comparison of spatial distribution of simulated mean daily AOD and
satellite-based as well as reanalyzed mean daily AOD from April 20 to May 10, 2016,
over the study area.

69

---

## Author Response (AR2)

1 Dear editor,

Thank you for your kind considerations on our manuscript entitled "Aerosol-2 meteorology feedback diminishes the trans-boundary transport of black carbon into the 3 Tibetan Plateau" (egusphere-2023-252). We appreciate that you gave us a chance of 4 minor revision to improve our manuscript to a level suitable for publication in ACP. We 5 6 also want to express our deep thanks to your positive comments. Those comments are 7 all valuable and very helpful for revising and improving our paper. We have studied comments carefully and have made corrections, which we hope meet with approval. 8 The main corrections in the paper and the corresponding responds are as following: 9

10

**11 Answers to editor:**

12 1. Both reviewers have viewed the manuscript positively and recommended its publication after various concerns are addressed. The authors have also responded to 13 the reviewers' comments and addressed their concerns in a satisfactory manner. 14 However, the authors' responses, including the requested explanations, justifications, 15 validations and supporting figures, are mostly contained within the response file itself 16 (showing that the authors have been engaged in a spirited conversation with the 17 18 reviewers there), but these changes are not actively incorporated into the revised manuscript or its supplement. As far as one can see, the revised manuscript contains a 19 20 lot fewer modified texts and figures than what the authors provided in the response file. The relevant changes are also not actively cited in the response file. The authors are 21 22 thus encouraged to actively incorporate the requested explanations, justifications, new validations, and their supporting figures (such as A1 and A2) into the revised 23 24 manuscript and its supplement, and cite them thoroughly in the response file. This is 25 necessary to create a standalone manuscript that fully and unambiguously addresses the reviewers' concerns without the need to refer back to the response file, which most 26 27 readers would not do.

Response: Firstly, we appreciate that you gave us a chance of minor revision to improve 28 our manuscript to a level suitable for publication in Atmospheric Chemistry and Physics. 29 We also want to express our deep thanks to your positive comments. According to your 30 valuable suggestion, we have actively incorporated the requested explanations, 31 justifications, new validations, and their supporting figures (such as A1 and A2) into 32 the revised manuscript and its supplement, and cite them thoroughly in the response 33 file. In addition, the answers to reviewers have also been made corresponding 34 modifications. 35

36

37 2. Moreover, the authors are recommended to further modify the concluding section according to ACP's writing guidelines (see the end of this response). In particular, the 38 authors are recommended to discuss the limitations and caveats of their study (currently 39 the last paragraph) in a tone suitable for a journal. Currently, the last paragraph, which 40 was written mostly to respond to the reviewers' suggestions, appears to belong more to 41 a research proposal than a research paper (e.g., "we plan to..."; "... will be used"). The 42 readers are generally not very interested in what the authors plan to do, but more in 43 what the limitations of the study mean for their interpretation of its results, and what 44

they themselves (i.e., the readers, the other researchers in general) should do in the future to address these limitations. Furthermore, the revised manuscript has not fully discussed the larger implications and importance of their results to the larger atmospheric/climate scientists' community, which is among the most important element of a good ACP paper.

50 Response: Thank you for your valuable suggestion. According to your suggestion, the last paragraph has been re-written as: Based on a severe aerosol pollution event, this 51 study investigates the potential impact of aerosol-meteorology feedback on the 52 transport of BC to the southern TP for a relatively short period. Similar studies for a 53 long-term period are necessary to examine whether the results obtained in this study are 54 universal. In addition, a finer grid resolution of the model domain and an improvement 55 in spatio-temporal resolution in emission inventory is needed to minimize the modeled 56 57 biases caused by the TP's particularly complex topography. Furthermore, the results of this study show for the first time that the aerosol-meteorology feedback plays a 58 substantial role in the transboundary transport of aerosols to the TP. Aerosols over the 59 TP exert an important effect on the convective system over the TP (Zhao et al., 60 2020;Zhou et al., 2017). In the monsoon season, the convective activity is very vigorous 61 62 over the TP. The transboundary transported aerosols along the slope from the foothill up to the TP via aerosol-meteorology feedback may also play a role. The potential 63 64 impacts of aerosols on the regional climate over the TP using a high-resolution model that can resolve the complex topography of the TP deserve in-depth investigation. 65

66

3. Finally, in the modified text of the abstract, it is not clear what "acquired" means;
they might have meant "determined" or "found" instead. That sentence also contains
too many repeated words; the whole first part of the sentence "By excavating the..." can
likely be omitted.

Response: Thank you for your good advice. The corresponding sentence in the abstract has been revised as 'In addition, it is found that the aerosol-meteorology feedback decreases the vertically integrated transboundary transport flux of BC from the central and western Himalayas towards the TP'.

75

76

**77 Answers to reviewer#1:**

1. The measurement distinguishes this manuscript and also a previous one, i.e., zhang et al., 2020, written by the key authors from a pure model simulation. However, these major results on transport flux and the aerosol-meteorology feedback are merely model simulations? Whether do the observation here or in other places reflect the pattern of transport flux or the aerosol-meteorology feedback pattern on transport flux?

Response: Thank you for your valuable suggestion. As the reviewer stated, the major
 results on transport flux and the aerosol-meteorology feedback are from model

simulations. Because known as the 'Third Pole', the Himalayas and the Tibetan Plateau

86 (TP) have very limited observational dataset due to harsh environment, limited access

87 for fieldwork, and the sparsity of fixed instrumental stations.

Through the literature research, it is found that there are studies using 88 observational dataset to reflect the transport flux of aerosols. For example, using 89 ground-based multi-axis differential optical absorption spectroscopy (MAX-DOAS) at 90 the Nancheng site in suburban Beijing on the southwest transport pathway of the 91 Beijing-Tianjin-Hebei (BTH) region, Hu et at. (2022) estimated the vertical profiles of 92 93 transport fluxes in the southwest-northeast direction. The results showed that the maximum net transport fluxes per unit cross-sectional area, calculated as pollutant 94 concentration multiply by wind speed, of aerosol extinction coefficient (AEC), NO2, 95 SO2 and HCHO were 0.98 km-1m s-1, 24, 14 and 8.0  $\mu$ g m-2 s-1 from southwest to 96 northeast, which occurred in the 200-300 m, 100-200 m, 500-600 m and 500-600 m 97 layers, respectively, due to much higher pollutant concentrations during southwest 98 99 transport than during northeast transport in these layers. The average net column transport fluxes were 1200 km-1 m2 s-1, 38, 26 and 15 mg m-1 s-1 from southwest to 100 northeast for AEC, NO2, SO2 and HCHO, respectively, in which the fluxes in the surface 101 layer (0-100 m) accounted for only 2.3%-4.2%. 102

However, in terms of the influence of aerosol-meteorology feedback on transport 103 flux of aerosols, it is found that no matter in regions with abundant observational data 104 105 or in regions with sparse observational data, the influence of aerosol-meteorology feedback on the transport flux of aerosols was evaluated by means of model simulation, 106 107 because sensitivity tests are involved in such studies. For instance, Huang et al. (2020) suggested that the aerosol-meteorology interaction and feedback enhanced the trans-108 109 boundary transport of pollutants between the North China Plain and the Yangzi River Delta regions and thus exacerbated the haze levels in these two regions simultaneously, 110 which was published in nature geoscience. 111

112

2. Although observation-model comparison on BC concentration and AOD have been
conducted, these results do not fully justify the simulated transport flux pattern? More
regious comparisons also transport flux among observations/reanalysis and models, or
intermodel comparison on transport flux, might be helpful?

Response: Thank you for your good advice. The reviewer made a very good point here. 117 According to the reviewer's suggestion, we not only validated the model performance 118 on temporal variation in AOD at different stations by comparing the simulated AOD 119 with the ground-based and satellite-based observational AOD (Figure S4), but also 120 verified the model performance on the spatial distribution of AOD over the study area 121 by comparing the simulated AOD with the satellite-based and reanalyzed AOD (Figure 122 123 S5). However, for BC, the comparison between the reanalysis and simulation was only conducted for most of the selected stations because we have very limited in-situ 124 125 observed BC (the observed BC is only available at the QOMS station). According to the reviewer's suggestion, we further validated the model performance by conducting 126 the inter-comparison among in-situ observation, simulation, and MERRA-2 reanalysis 127 at the QOMS station, as shown in Figure S6. The results show that the temporal 128 variation in simulation is very close to that of simulation. Moreover, the correlation 129 coefficient between the simulation and observation is 0.867, passing the 99% 130

confidence level. Therefore, the model configuration used in this study presents a 131 reasonable performance on BC. For the spatial distribution of BC over the study area, 132 because the spatial distribution of BC concentration retrieved from CAM Chem dataset 133 is not reasonable (Figure S7a), the inter-comparison between WRF-Chem simulated 134 and reanalyzed BC concentrations over the domain was performed to validate the model 135 136 performance on the spatial distribution of BC concentrations. The results show that the spatial pattern of WRF-Chem simulated BC is similar to that of MERRA-2 reanalyzed 137 BC (Figure S7b-c). 138

The section of validating the model performance on AOD and BC has been revised 139 as follows: 140

141 To validate the model performance on simulating spatiotemporal variations in 142 aerosols, ground-based AOD from AERONET together with reanalyzed AOD from 143 MERRA-2 is compared to simulated AOD first. Figure S4 shows the temporal variations in simulated and observed daily mean AOD at Nam Co, QOMS, and Pokhara 144 stations for the period from April 20 to May 10, 2016. As a whole, the WRF-Chem 145 model reasonably reproduces the temporal variations in AOD at each of the above 146 stations, with a relatively larger bias at Nam Co and Pokhara stations and a smaller bias 147 148 at QOMS station. The specific statistics for N, observed mean, simulated mean, MB, NMB, RMSE, and R between observed and simulated AOD at different stations are 149 150 shown in Table S2 in the SI. As we note that one-third of AOD values at the selected stations in the MERRA-2 dataset during the study period is missing, the statistical 151 description between the reanalyzed and simulated AOD is not presented. The results 152 from Table S2 indicate that MB with values of -0.13, -0.01, and -0.57, and R with 153 154 values of 0.58, 0.42, and 0.56 are obtained at Nam Co, QOMS, and Pokhara, respectively. Moreover, AOD from observation is significantly correlated with that 155 from simulation at Nam Co and Pokhara stations, with the correlation coefficient 156 157 passing the 95% confidence level. In addition, we note that AOD from the simulation is on average lower than that from observation, which may be due to the assumed 158 spherical aerosol particles in the model simulation. The optical properties of particles 159 are more sensitive to non-spherical morphology than primary spherical structure (China 160 et al., 2015;He et al., 2015). On the whole, the model effectively reproduces the 161 observed temporal variation in AOD. 162

Spatially, the spatial pattern of simulated AOD is in consistent with that from either 163 MODIS or MERRA-2 reanalysis dataset, suggesting a reliable performance of WRF-164 Chem on simulating AOD. Specifically, AOD from simulation, MODIS, and MERRA-165 166 2 shows distinct spatial distribution characteristics, with high values in northern South Asia, the Bay of Bengal, Southeast Asia, and the Sichuan Basin and low values over 167 the TP (Figure S5). This is because northern South Asia, Southeast Asia, and the 168 Sichuan Basin are heavily industrialized and densely populated regions compared to 169 the TP (Bran and Srivastava, 2017). In the Taklimakan Desert, AOD monitored by 170 satellite is much higher than that obtained from simulation, which is likely due to the 171 172 uncertainty of the emission inventory. Taken together, the comparison between simulation from WRF-Chem and observation from AERONET, MODIS, and MERRA-173

174 2 shows that the WRF-Chem model captures the overall spatio-temporal characteristics175 of AOD over the domain.

- To verify the capability of this framework of WRF-Chem on simulating BC 176 concentration, we present the temporal variation in simulated and reanalyzed hourly 177 BC concentrations at Nam Co, OOMS, Lhasa, NCO-P, Laohugou, and Kanpur stations 178 179 during the period from April 20 to May 10, 2016, as shown in Figure 4. It is found that the WRF-Chem model overall reproduces the temporal variation in reanalyzed BC 180 concentrations at different stations. The specific statistics for N, observed mean, 181 simulated mean, MB, NMB, RMSE, and R between the reanalyzed and simulated 182 hourly BC concentrations at different stations are shown in Table S2 in the SI. As can 183 be seen, MB with values of -0.07, 0.14, -0.02, -0.02, 0.02, and 0.72, and R with values 184 185 of 0.67, 0.43, 0.47, 0.50, 0.25, and 0.64 are obtained at Nam Co, QOMS, Lhasa, NCO-186 P, Laohugou, and Kanpur stations, respectively. The reanalyzed and simulated hourly BC concentrations are strongly correlated at each of the stations, with the correlation 187 coefficient exceeding the 99% confidence level. Besides the reanalyzed hourly BC 188 concentration, the in-situ BC observation is available at the OOMS station. The inter-189 comparison among in-situ observed, simulated, and MERRA-2 reanalyzed daily mean 190 191 BC concentrations at the QOMS station was further conducted, as shown in Figure S6. It is apparent that the temporal variation in simulated daily mean BC concentrations is 192 193 very close to that of observed daily mean BC concentrations. The correlation coefficient between the simulated and observed daily mean BC concentrations is 0.867, passing 194 195 the 99% confidence level. Hence, the WRF-Chem model exhibits a better performance in simulating BC concentrations. 196
- 197 Because the spatial distribution of BC concentration retrieved from CAM Chem 198 dataset is not reasonable (Figure S7a), the inter-comparison between WRF-Chem 199 simulated and reanalyzed BC concentrations over the domain was performed to validate 200 the model performance on the spatial distribution of BC concentrations. Figure S7b-c presents the spatial distribution of simulated and reanalyzed BC concentrations over the 201 domain averaged for the period from from April 20 to May 10, 2016. It can be found 202 that BC concentrations from both simulation and reanalysis display distinct spatial 203 variability, with low concentrations over the TP and high concentrations over the north 204 of South Asia, Southeast Asia, and the Sichuan Basin. As one of the most pristine 205 regions on the earth, the TP has a small population density and a low degree of 206 industrialization, resulting in low BC concentrations. Nonetheless, regions adjacent to 207 the TP like north of South Asia, Southeast Asia, and the Sichuan Basin with low 208 209 elevations have dense populations and developed industrialization (Li et al., 2016a;Qin and Xie, 2012;Li et al., 2016b), emitting large amounts of BC into the atmosphere and 210 resulting in high BC concentrations. Therefore, the WRF-Chem model can capture the 211 main temporal and spatial features of BC concentrations over the TP and adjacent 212 regions. 213

215 Figure 4. Temporal variations in simulated and reanalyzed hourly BC concentrations at

216 Nam Co (a), QOMS (b), Lhasa (c), NCO-P (d), Laohugou (e), and Kanpur (f) stations

- for the period from April 20 to May 10, 2016.
- 218

219

Figure S4. Inter-comparison of temporal variations in simulated, ground-based, and satellite-based daily mean AOD at (a) Nam Co, (b) QOMS, and (c) Pokhara stations for the period from April 20 to May 10, 2016.

223